# Detecting and Adapting to Irregular Distribution Shifts in Bayesian Online Learning

**Aodong Li[1]    Alex Boyd[2]    Padhraic Smyth[1,2]    Stephan Mandt[1,2]**

[1]Department of Computer Science    [2]Department of Statistics
University of California, Irvine
{aodongl1, alexjb, mandt}@uci.edu    smyth@ics.uci.edu

## Abstract

We consider the problem of online learning in the presence of distribution shifts that occur at an unknown rate and of unknown intensity. We derive a new Bayesian online inference approach to simultaneously infer these distribution shifts and adapt the model to the detected changes by integrating ideas from change point detection, switching dynamical systems, and Bayesian online learning. Using a binary 'change variable,' we construct an informative prior such that–if a change is detected–the model partially erases the information of past model updates by tempering to facilitate adaptation to the new data distribution. Furthermore, the approach uses beam search to track multiple change-point hypotheses and selects the most probable one in hindsight. Our proposed method is model-agnostic, applicable in both supervised and unsupervised learning settings, suitable for an environment of concept drifts or covariate drifts, and yields improvements over state-of-the-art Bayesian online learning approaches.

## 1  Introduction

Deployed machine learning systems are often faced with the problem of distribution shift, where the new data that the model processes is systematically different from the data the system was trained on [Zech et al., 2018, Ovadia et al., 2019]. Furthermore, a shift can happen anytime after deployment, unbeknownst to the users, with dramatic consequences for systems such as self-driving cars, robots, and financial trading algorithms, among many other examples.

Updating a deployed model on new, representative data can help mitigate these issues and improve general performance in most cases. This task is commonly referred to as *online* or *incremental learning*. Such online learning algorithms face a tradeoff between remembering and adapting. If they adapt too fast, their performance will suffer since adaptation usually implies that the model loses memory of previously encountered training data (which may still be relevant to future predictions). On the other hand, if a model remembers too much, it typically has problems adapting to new data distributions due to its finite capacity.

The tradeoff between adapting and remembering can be elegantly formalized in a Bayesian online learning framework, where a prior distribution is used to keep track of previously learned parameter estimates and their confidences. For instance, variational continual learning (VCL) [Nguyen et al., 2018] is a popular framework that uses a model's previous posterior distribution as the prior for new data. However, the assumption of such continual learning setups is usually that the data distribution is stationary and not subject to change, in which case adaptation is not an issue.

This paper proposes a new Bayesian online learning framework suitable for non-stationary data distributions. It is based on two assumptions: (i) distribution shifts occur irregularly and must be inferred from the data, and (ii) the model requires not only a good mechanism to aggregate data but

35th Conference on Neural Information Processing Systems (NeurIPS 2021).

also the ability to partially forget information that has become obsolete. To solve both problems, we still use a Bayesian framework for online learning (i.e., letting a previous posterior distribution inform the next prior); however, before combining the previously learned posterior with new data evidence, we introduce an intermediate step. This step allows the model to either broaden the previous posterior's variance to reduce the model's confidence, thus providing more "room" for new information, or remain in the same state (i.e., retain the unchanged, last posterior as the new prior).

We propose a mechanism for enabling this decision by introducing a discrete "change variable" that indicates the model's best estimate of whether the data in the new batch is compatible with the previous data distribution or not; the outcome then informs the Bayesian prior at the next time step. We further augment the scheme by performing beam search on the change variable. This way, we are integrating change detection and Bayesian online learning into a common framework.

We test our framework on a variety of real-world datasets that show concept drift, including basketball player trajectories, malware characteristics, sensor data, and electricity prices. We also study sequential versions of SVHN and CIFAR-10 with covariate drift, where we simulate the shifts in terms of image rotations. Finally, we study word embedding dynamics in an unsupervised learning approach. Our approach leads to a more compact and interpretable latent structure and significantly improved performance in the supervised experiments. Furthermore, it is highly scalable; we demonstrate it on models with hundreds of thousands of parameters and tens of thousands of feature dimensions.

Our paper is structured as follows: we review related work in Section 2, introduce our methods in Section 3, report our experiments in Section 4, and draw conclusions in Section 6.

## 2 Related Work

Our paper connects to Bayesian online learning, change detection, and switching dynamical systems.

**Bayesian Online and Continual Learning**  There is a rich existing literature on Bayesian and continual learning. The main challenge in streaming setups is to reduce the impact of old data on the model which can be done by exponentially decaying old samples [Honkela and Valpola, 2003, Sato, 2001, Graepel et al., 2010] or re-weighting them [McInerney et al., 2015, Theis and Hoffman, 2015]. An alternative approach is to adapt the model posterior between time steps, such as tempering it at a constant rate to accommodate new information [Kulhavỳ and Zarrop, 1993, Kurle et al., 2020]. In contrast, *continual learning* typically assumes a stationary data distribution and simply uses the old posterior as the new prior. A scalable such scheme based on variational inference was proposed by [Nguyen et al., 2018] which was extended by several authors [Farquhar and Gal, 2018, Schwarz et al., 2018]. A related concept is elastic weight consolidation [Kirkpatrick et al., 2017], where new model parameters are regularized towards old parameter values.

All of these approaches need to make assumptions on the expected frequency and strengh of change which are hard-coded in the model parameters (e.g., exponential decay rates, re-weighting terms, prior strengths, or temperature choices). Our approach, in contrast, detects change based on a discrete variable and makes no assumption about its frequency. Other approaches assume situations where data arrive in irregular time intervals, but are still concerned with static data distributions [Titsias et al., 2019, Lee et al., 2020, Rao et al., 2019].

**Change Point Models**  There is also a rich history of models for change detection. A popular class of change point models includes "product partition models" [Barry and Hartigan, 1992] which assume independence of the data distribution across segments. In this regime, Fearnhead [2005] proposed change detection in the context of regression and generalized it to online inference [Fearnhead and Liu, 2007]; Adams and MacKay [2007] described a Bayesian *online* change point detection scheme (BOCD) based on conditional conjugacy assumptions for one-dimensional sequences. Other work generalized change detection algorithms to multivariate time series [Xuan and Murphy, 2007, Xie et al., 2012] and non-conjugate Bayesian inference [Saatçi et al., 2010, Knoblauch and Damoulas, 2018, Turner et al., 2013, Knoblauch et al., 2018].

Our approach relies on jointly inferring changes in the data distribution while carrying out Bayesian parameter updates for adaptation. To this end, we detect change in the high-dimensional space of

model (e.g., neural network) parameters, as opposed to directly in the data space. Furthermore, a detected change only *partially* resets the model parameters, as opposed to triggering a complete reset.

Titsias et al. [2020] proposed change detection to detect distribution shifts in sequences based on low-dimensional summary statistics such as a loss function; however, the proposed framework does not use an informative prior but requires complete retraining.

**Switching Linear Dynamical Systems**    Since our approach integrates a discrete change variable, it is also loosely connected to the topic of switching linear dynamical systems. Linderman et al. [2017] considered *recurrent* switching linear dynamical systems, relying on Bayesian conjugacy and closed-form message passing updates. Becker-Ehmck et al. [2019] proposed a variational Bayes filtering framework for switching linear dynamical systems. Murphy [2012] and Barber [2012] developed an inference method using a Gaussian sum filter. Instead, we focus on inferring the full history of discrete latent variable values instead of just the most recent one.

Bracegirdle and Barber [2011] introduce a *reset* variable that sets the continuous latent variable to an unconditional prior. It is similar to our work, but relies on using low-dimensional, tractable models. Our tempering prior can be seen as a partial reset, augmented with beam search. We also extend the scope of switching dynamical systems by integrating them into a supervised learning framework.

## 3   Methods

**Overview**    Section 3.1 introduces the setup and the novel model structure under consideration. Section 3.2 introduces an exact inference scheme based on beam search. Finally, we introduce the variational inference extension for intractable likelihood models in Section 3.3.

### 3.1   Problem Assumptions and Structure

We consider a stream of data that arrives in batches $\mathbf{x}_t$ at discrete times $t$.[1] For supervised setups, we consider pairs of features and targets $(\mathbf{x}_t, \mathbf{y}_t)$, where the task is to model $p(\mathbf{y}_t|\mathbf{x}_t)$. An example model could be a Bayesian neural network, and the parameters $\mathbf{z}_t$ could be the network weights. For notational simplicity we focus on the unsupervised case, where the task is to model $p(\mathbf{x}_t)$ using a model $p(\mathbf{x}_t|\mathbf{z}_t)$ with parameters $\mathbf{z}_t$ that we would like to tune to each new batch.[2] We then measure the prediction error either on one-step-ahead samples or using a held-out test set.

Furthermore, we assume that while the $\mathbf{x}_t$ are i.i.d. within batches, they are not necessarily i.i.d. across batches as they come from a time-varying distribution $p_t(\mathbf{x}_t)$ (or $p_t(\mathbf{x}_t, \mathbf{y}_t)$ in the supervised cases) which is subject to distribution shifts. We do not assume whether these distribution shifts occur instantaneously or gradually. The challenge is to optimally adapt the parameters $\mathbf{z}_t$ to each new batch while borrowing statistical strength from previous batches.

As follows, we will construct a Bayesian online learning scheme that accounts for changes in the data distribution. For every new batch of data, our scheme tests whether the new batch is compatible with the old data distribution, or more plausible under the assumption of a change. To this end, we employ a binary "change variable" $s_t$, with $s_t = 0$ for no detected change and $s_t = 1$ for a detected change. Our model's joint distribution factorizes as follows:

$$p(\mathbf{x}_{1:T}, \mathbf{z}_{1:T}, s_{1:T}) = \prod_{t=1}^{T} p(\mathbf{x}_t|\mathbf{z}_t)p(\mathbf{z}_t|s_t; \boldsymbol{\tau}_t)p(s_t). \tag{1}$$

We assumed a factorized Bernoulli prior $\prod_t p(s_t)$ over the change variable: an assumption that will simplify the inference, but which can be relaxed. As a result, our model is fully-factorized over time, however, the model can still capture temporal dependencies through the informative prior $p(\mathbf{z}_t|s_t; \boldsymbol{\tau}_t)$. Temporal information enters this prior through certain *sufficient statistics* $\boldsymbol{\tau}_t$ that depend on properties of the previous time-step's approximate posterior.

In more detail, $\boldsymbol{\tau}_t$ is a *functional* on the previous time step's approximate posterior, $\boldsymbol{\tau}_t = \mathcal{F}[p(\mathbf{z}_{t-1}|\mathbf{x}_{1:t-1}, s_{1:t-1})]$.[3]  Throughout this paper, we will use a specific form of $\boldsymbol{\tau}_t$, namely

---

[1]In an extreme case, it is possible for a batch to include only a single data point.

[2]In supervised setups, we consider a conditional model $p(\mathbf{y}_t|\mathbf{z}_t, \mathbf{x}_t)$ with features $\mathbf{x}_t$ and targets $\mathbf{y}_t$.

[3]Subscripts $1:t-1$ indicates the integers from 1 to $t-1$ inclusively.

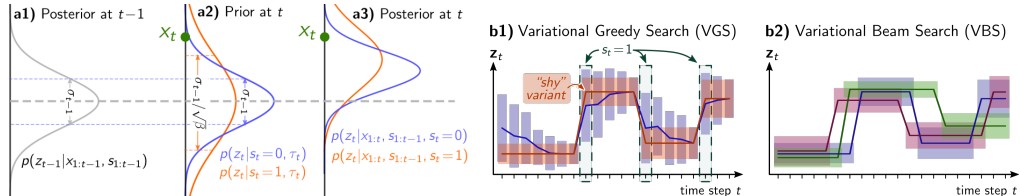

Figure 1: **a)** A single inference step for the latent mean in a 1D linear Gaussian model. Starting from the previous posterior (**a1**), we consider both its broadened and un-broadened version (**a2**). Then the model absorbs the observation and updates the priors (**a3**). **b)** Sparse inference via greedy search (**b1**) and variational beam search (**b2**). b) Solid lines indicate fitted mean $\mu_t$ over time steps $t$ with boxes representing $\pm 1\sigma$ error bars. See more information about the pictured "shy" variant in Supplement C.

capturing the previous posterior's mean and variance.[4] More formally,

$$\boldsymbol{\tau}_t \equiv \{\boldsymbol{\mu}_{t-1}, \Sigma_{t-1}\} \equiv \{\text{Mean}, \text{Var}\}[\mathbf{z}_{t-1}|\mathbf{x}_{1:t-1}, s_{1:t-1}]. \tag{2}$$

Based on this choice, we define the conditional prior as follows:

$$p(\mathbf{z}_t|s_t; \boldsymbol{\tau}_t) = \begin{cases} \mathcal{N}(\mathbf{z}_t; \boldsymbol{\mu}_{t-1}, \Sigma_{t-1}) & \text{for } s_t = 0 \\ \mathcal{N}(\mathbf{z}_t; \boldsymbol{\mu}_{t-1}, \beta^{-1}\Sigma_{t-1}) & \text{for } s_t = 1 \end{cases} \tag{3}$$

Above, $0 < \beta < 1$ is a hyperparameter referred to as *inverse temperature*[5]. If no change is detected (i.e., $s_t = 0$), our prior becomes a Gaussian distribution centered around the previous posterior's mean and variance. In particular, if the previous posterior was already Gaussian, it becomes the new prior. In contrast, if a change was detected, the *broadened* posterior becomes the new prior.

For as long as no changes are detected ($s_t = 0$), the procedure results in a simple Bayesian online learning procedure, where the posterior uncertainty shrinks with every new observation. In contrast, if a change is detected ($s_t = 1$), an overconfident prior would be harmful for learning as the model needs to adapt to the new data distribution. We therefore weaken the prior through *tempering*. Given a temperature $\beta$, we raise the previous posterior's Gaussian approximation to the power $\beta$, renormalize it, and then use it as a prior for the current time step.

The process of tempering the Gaussian posterior approximation can be understood as removing equal amounts of information in any direction in the latent space. To see this, let $\mathbf{z}$ be a multivariate Gaussian with covariance $\Sigma$ and $\mathbf{u}$ be a unit direction vector. Then tempering removes an equal amount of information regardless of $\mathbf{u}$, $H_{\mathbf{u}} = \frac{1}{2}\log(2\pi e \mathbf{u}^\top \Sigma \mathbf{u}) - \frac{1}{2}\log\beta$, erasing learned information to free-up model capacity to adjust to the new data distribution. See Supplement B for more details.

**Connection to Sequence Modeling**   Our model assumptions have a resemblance to time series modeling: if we replaced $\boldsymbol{\tau}_t$ with $\mathbf{z}_{t-1}$, we would condition on previous latent states rather than posterior summary statistics. In contrast, our model still factorizes over time and therefore makes weaker assumptions on temporal continuity. Rather than imposing temporal continuity on a data *instance* level, we instead assume temporal continuity at a *distribution* level.

**Connection to Changepoint Modeling.**   We also share similar assumptions with the changepoint literature [Barry and Hartigan, 1992]. However, in most cases, these models don't assume an informative prior, effectively not taking into account any sufficient statistics $\boldsymbol{\tau}_t$. This forces these models to re-learn model parameters from scratch after a detected change, whereas our approach allows for some transfer of information before and after the distribution shift.

## 3.2   Exact Inference

Before presenting a scalable variational inference scheme in our model, we describe an exact inference scheme when everything is tractable, i.e., the special case of linear Gaussian models.

---

[4]In later sections, we will use a Gaussian approximation to the posterior, but here it is enough to assume that these quantities are computable.

[5]In general it only requires $\beta > 0$ to be inverse temperature. We further assume $\beta < 1$ in this paper as this value interval broadens and weakens the previous posterior. See the following paragraphs.

According to our assumptions, the distribution shifts occur at discrete times and are unobserved. Therefore, we have to infer them from the observed data and adapt the model accordingly. Recall the distribution shift is represented by the binary latent variable $s_t$ at time step $t$. Inferring the posterior over $s_t$ at $t$ will thus notify us how likely the change happens under the model assumption. As follows, we show the posterior of $s_t$ is simple in a tractable model and bears similarity with a likelihood ratio test. Suppose we moved from time step $t-1$ to step $t$ and observed new data $\mathbf{x}_t$. Denote the history decisions and observations by $\{s_{1:t-1}, \mathbf{x}_{1:t-1}\}$, which enters through $\boldsymbol{\tau}_t$. Then by Bayes rule, the exact posterior over $s_t$ is again a Bernoulli, $p(s_t|s_{1:t-1}, \mathbf{x}_{1:t}) = \text{Bern}(s_t; m)$, with parameter

$$m = \sigma\left(\log \frac{p(\mathbf{x}_t|s_t=1, s_{1:t-1}, \mathbf{x}_{1:t-1})p(s_t=1)}{p(\mathbf{x}_t|s_t=0, s_{1:t-1}, \mathbf{x}_{1:t-1})p(s_t=0)}\right) = \sigma\left(\log \frac{p(\mathbf{x}_t|s_t=1, s_{1:t-1}, \mathbf{x}_{1:t-1})}{p(\mathbf{x}_t|s_t=0, s_{1:t-1}, \mathbf{x}_{1:t-1})} + \xi_0\right). \tag{4}$$

Above, $\sigma$ is the sigmoid function, and $\xi_0 = \log p(s_t=1) - \log p(s_t=0)$ are the log-odds of the prior $p(s_t)$ and serves as a bias term. $p(\mathbf{x}_t|s_{1:t}, \mathbf{x}_{1:t-1}) = \int p(\mathbf{x}_t|\mathbf{z}_t)p(\mathbf{z}_t|s_t; \boldsymbol{\tau}_t)d\mathbf{z}_t$ is the model evidence. Overall, $m$ specifies the probability of $s_t=1$ given $\mathbf{x}_{1:t}$ and $s_{1:t-1}$.

Eq. 4 has a simple interpretation as a likelihood ratio test: a change is more or less likely depending on whether or not the observations $\mathbf{x}_t$ are better explained under the assumption of a detected change.

We have described the detection procedure thus far, now we turn to the adaptation procedure. To adjust the model parameters $\mathbf{z}_t$ to the new data given a change or not, we combine the likelihood of $\mathbf{x}_t$ with the conditional prior (Eq. 3). This corresponds to the posterior of $\mathbf{z}_t$, $p(\mathbf{z}_t|\mathbf{x}_{1:t}, s_{1:t}) = \frac{p(\mathbf{x}_t|\mathbf{z}_t)p(\mathbf{z}_t|s_t; \boldsymbol{\tau}_t)}{p(\mathbf{x}_t|s_{1:t}, \mathbf{x}_{1:t-1})}$, obtained by Bayes rule. The adaptation procedure is illustrated in Fig. 1 (a), where we show how a new observation modifies the conditional prior of model parameters.

As a result of Eq. 4, we obtain the marginal distribution of $\mathbf{z}_t$ at time $t$ as a binary mixture with mixture weights $p(s_t=1|s_{1:t-1}, \mathbf{x}_{1:t}) = m$ and $p(s_t=0|s_{1:t-1}, \mathbf{x}_{1:t}) = 1-m$: $p(\mathbf{z}_t|s_{1:t-1}, \mathbf{x}_{1:t}) = mp(\mathbf{z}_t|s_t=1, s_{1:t-1}, \mathbf{x}_{1:t}) + (1-m)p(\mathbf{z}_t|s_t=0, s_{1:t-1}, \mathbf{x}_{1:t})$.

**Exponential Branching**  We note that while we had originally started with a posterior $p(\mathbf{z}_{t-1}|\mathbf{x}_{1:t-1}, s_{1:t-1})$ at the previous time, our inference scheme resulted in $p(\mathbf{z}_t|s_{1:t-1}, \mathbf{x}_{1:t})$ being a mixture of two components as it branches over two possible states.[6] When we iterate, we encounter an exponential branching of possibilities, or *hypotheses* over possible sequences of regime shifts $s_{1:t}$. To still carry out the filtering scheme efficiently, we need a truncation scheme, e.g., approximate the bimodal marginal distribution by a unimodal one. As follows, we will discuss two methods—greedy search and beam search—to achieve this goal.

**Greedy Search**  In the simplest "greedy" setup, we train the model in an online fashion by iterating over time steps $t$. For each $t$, we update a *truncated* distribution via the following three steps:

1. Compute the conditional prior $p(\mathbf{z}_t|s_t; \boldsymbol{\tau}_t)$ (Eq. 3) based on $p(\mathbf{z}_{t-1}|\mathbf{x}_{1:t-1}, s_{1:t-1})$ and evaluate the likelihood $p(\mathbf{x}_t|\mathbf{z}_t)$ upon observing data $\mathbf{x}_t$.

2. Infer whether a change happens or not using the posterior over $s_t$ (Eq. 4) and adapt the model parameters $\mathbf{z}_t$ for each case.

3. Select $s_t \in \{0, 1\}$ that has larger posterior probability $p(s_t|s_{1:t-1}, \mathbf{x}_{1:t})$ and its corresponding model hypothesis $p(\mathbf{z}_t|s_{1:t}, \mathbf{x}_{1:t})$ (i.e., make a "hard" decision over $s_t$ with a threshold of $\frac{1}{2}$).

The above filtering algorithm iteratively updates the posterior distribution over $\mathbf{z}_t$ each time it observes new data $\mathbf{x}_t$. In the version of greedy search discussed above, the approach decides immediately, i.e., before observing subsequent data points, whether a change in $\mathbf{z}_t$ has occurred or not in step 3. (Please note the decision is irrelevant to history, as opposed to the beam search described below.) We illustrate greedy search is illustrated in Fig. 1 (b1) where VGS is the variational inference counterpart.

**Beam Search**  A greedy search is prone to missing change points in data sets with a low signal/noise ratio per time step because it cannot accumulate evidence for a change point over a series of time steps. The most obvious improvement over greedy search that has the ability to accumulate evidence for a change point is beam search. Rather than deciding greedily whether a change occurred or not at each time step, beam search considers both cases in parallel as it delays the decision of which

---

[6]See also Fig. 1 in Supplement C.

one is more likely (see Fig. 1 (b2) and Fig. 2 (left) for illustration). The algorithm keeps track of a fixed number $K > 1$ of possible hypotheses of change points. For each hypothesis, it iteratively updates the posterior distribution as a greedy search. At time step $t$, every potential continuation of the $K$ sequences is considered with $s_t \in \{0, 1\}$, thus doubling the number of histories of which the algorithm has to track. To keep the computational requirements bounded, beam search thus discards half of the sequences based on an exploration-exploitation trade-off.

Beam search simultaneously tracks multiple hypotheses necessitating the differentiation between them. In greedy search, we can distinguish hypotheses based on the most recent $s_t$'s value since only two hypotheses are considered at each step. However, beam search considers at most $2K$ hypotheses each step, which exceeds the capacity of a single $s_t$. We thus resort to the decision history $s_{1:t-1}$ to further tell hypotheses apart. The weight $p(s_{1:t}|\mathbf{x}_{1:t})$ of each hypothesis can be computed recursively:

$$
\begin{aligned}
p(s_{1:t}|\mathbf{x}_{1:t}) &\propto p(s_t, \mathbf{x}_t|s_{1:t-1}, \mathbf{x}_{1:t-1})p(s_{1:t-1}|\mathbf{x}_{1:t-1}) \\
&\propto p(s_t|s_{1:t-1}, \mathbf{x}_{1:t})p(s_{1:t-1}|\mathbf{x}_{1:t-1})
\end{aligned}
\tag{5}
$$

where the added information $p(s_t|s_{1:t-1}, \mathbf{x}_{1:t})$ at step $t$ is the posterior of $s_t$ (Eq. 4). This suggests the "correction in hindsight" nature of beam search: re-ranking the sequence $s_{1:t}$ as a whole at time $t$ indicates the ability to correct decisions before time $t$.

Another ingredient is a set $\mathbb{B}_t$, which contains the $K$ most probable "histories" $s_{1:t}$ at time $t$. From time $t - 1$ to $t$, we evaluate the continuation of each hypothesis $s_{1:t-1} \in \mathbb{B}_{t-1}$ as the first two steps of greedy search, leading to $2K$ hypotheses. We then compute the weight of each hypothesis using Eq. 5. Finally, select top $K$ hypotheses into $\mathbb{B}_t$ and re-normalize the weights of hypotheses in $\mathbb{B}_t$.

This concludes the recursion from time $t - 1$ to $t$. With $p(\mathbf{z}_t|s_{1:t}, \mathbf{x}_{1:t})$ and $p(s_{1:t}|\mathbf{x}_{1:t})$, we can achieve any marginal distribution of $\mathbf{z}_t$, such as $p(\mathbf{z}_t|\mathbf{x}_{1:t}) = \sum_{s_{1:t}} p(\mathbf{z}_t|s_{1:t}, \mathbf{x}_{1:t})p(s_{1:t}|\mathbf{x}_{1:t})$.

**Beam Search Diversification** Empirically, we find that the naive beam search procedure does not realize its full potential. As commonly encountered in beam search, histories over change points are largely shared among all members of the beam. To encourage diverse beams, we constructed the following simple scheme. While transitioning from time $t-1$ to $t$, every hypothesis splits into two scenarios, one with $s_t = 0$ and one with $s_t = 1$, resulting in $2K$ temporary hypotheses. If two resulting hypotheses only differ in their most recent $s_t$-value, we say that they come from the same "family." Each member among the $2K$ hypotheses is ranked according to its posterior probability $p(s_{1:t}|\mathbf{x}_{1:t})$ in Eq. 5. In a first step, we discard the bottom $1/3$ of the $2K$ hypotheses, leaving $4/3K$ hypotheses (we always take integer multiples of 3 for $K$). To truncate the beam size from $4/3K$ down to $K$, we rank the remaining hypotheses according to their posterior probability and pick the top $K$ ones while *also* ensuring that we pick a member from every remaining family. The diversification scheme ensures that underperforming families can survive, leading to a more diverse set of hypotheses. We found this beam diversification scheme to work robustly across a variety of experiments.

### 3.3 Variational Inference

In most practical applications, the evidence term is not available in closed-form, leaving Eq. 4 intractable to evaluate. However, we can follow a structured variational inference approach [Wainwright and Jordan, 2008, Hoffman and Blei, 2015, Zhang et al., 2018], defining a joint variational distribution $q(\mathbf{z}_t, s_t|s_{1:t-1}) = q(s_t|s_{1:t-1})q(\mathbf{z}_t|s_{1:t})$, to approximate $p(\mathbf{z}_t, s_t|s_{1:t-1}, \mathbf{x}_{1:t}) = p(s_t|s_{1:t-1}, \mathbf{x}_{1:t})p(\mathbf{z}_t|s_{1:t}, \mathbf{x}_{1:t})$. This procedure completes the detection and adaptation altogether.

One may wonder how the exact inference schemes for $s_t$ and $\mathbf{z}_t$ are modified in the structured variational inference scenario. In Supplement A, we derive the solution for $q(\mathbf{z}_t, s_t|s_{1:t-1})$. Surprisingly we have the following closed-form update equation for $q(s_t|s_{1:t-1})$ that bears strong similarities to Eq. 4. The new scheme simply replaces the intractable evidence term with a lower bound proxy – optimized conditional evidence lower bound $\mathcal{L}(q^*|s_{1:t})$ (CELBO, defined later), giving the update

$$
q^*(s_t|s_{1:t-1}) = \text{Bern}(s_t; m); \quad m = \sigma\left(\tfrac{1}{T}\mathcal{L}(q^*|s_t = 1, s_{1:t-1}) - \tfrac{1}{T}\mathcal{L}(q^*|s_t = 0, s_{1:t-1}) + \xi_0\right).
\tag{6}
$$

Above, we introduced a parameter $T \geq 1$ (not to be confused with $\beta$) to optionally downweigh the data evidence relative to the prior (see Experiments Section 4).

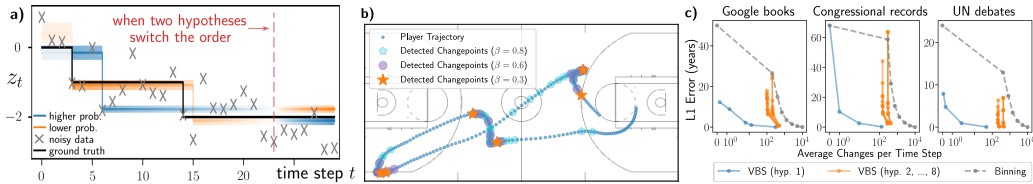

Figure 2: **a)** Inferring the mean (black line) of a time-varying data distribution (black samples) with VBS. The initially unlikely hypothesis begins dominating over the other at step 23. **b)** Basketball player tracking: ablation study over $\beta$ for VBS while fixing other parameters. We used greedy search (K=1) and run the model under different $\beta$ values. Increasing $\beta$ leads to more sensitivity to changes in data, leading to more detected changepoints. **c)** Document dating error as a function of model sparsity, measured in average words update per year. As semantic changes get successively sparsified by varying $\xi_0$ (Eq. 6), VBS maintains a better document dating performance compared to baselines.

Now we define the CELBO. To approximate $p(\mathbf{z}_t|s_{1:t}, \mathbf{x}_{1:t})$ by variational distribution $q(\mathbf{z}_t|s_{1:t})$, we minimize the KL divergence between $q(\mathbf{z}_t|s_{1:t})$ and $p(\mathbf{z}_t|s_{1:t}, \mathbf{x}_{1:t})$, leading to

$$q^*(\mathbf{z}_t|s_{1:t}) = \underset{q(\mathbf{z}_t|s_{1:t}) \in Q}{\arg\max} \; \mathcal{L}(q|s_{1:t}), \tag{7}$$

$$\mathcal{L}(q|s_{1:t}) := \mathbb{E}_q[\log p(\mathbf{x}_t|\mathbf{z}_t)] - \mathrm{KL}(q(\mathbf{z}_t|s_{1:t})||p(\mathbf{z}_t|s_t; \boldsymbol{\tau}_t)).$$

$Q$ denotes the variational family (i.e., factorized normal distributions), and we term $\mathcal{L}(q|s_{1:t})$ CELBO.

The greedy search and beam search schemes also apply to variational inference. We name them *variational greedy search* (VGS, VBS (K=1)) and *variational beam search* (VBS) (Fig. 1 (b)).

**Algorithm Complexity**   VBS's computational time and space complexity scale *linearly* with the beam size $K$. As such, its computational cost is only about $2K$ times larger than greedy search[7]. Furthermore, our algorithm's complexity is $O(1)$ in the sequence length $t$. It is not necessary to store sequences $s_{1:t}$ as they are just symbols to distinguish hypotheses. The only exception to this scaling would be an application asking for the most likely changepoint sequence in hindsight. In this case, the changepoint sequence (but not the associated model parameters) would need to be stored, incurring a cost of storing exactly $K \times T$ binary variables. This storage is, however, not necessary when the focus is only on adapting to distribution shifts.

## 4   Experiments

**Overview**   The objective of our experiments is to show that, compared to other methods, variational beam search (1) better reacts to different distribution shifts, e.g., *concept drifts* and *covariate drifts*, while (2) revealing interpretable and temporally sparse latent structure. We experiment on artificial data to demonstrate the "correct in hindsight" nature of VBS (Section 4.1), evaluate online linear regression on three datasets with concept shifts (Section 4.3), visualize the detected change points on basketball player movements, demonstrate the robustness of the hyperparameter $\beta$ (Section 4.3), study Bayesian deep learning approaches on sequences of transformed images with covariate shifts (Section 4.4), and study the dynamics of word embeddings on historical text corpora (Section 4.5). Unstated experimental details are in Supplement G.

### 4.1   An Illustrative Example

We first aim to dynamically demonstrate the "correction in hindsight" nature of VBS based on a simple setup involving artificial data. To this end, we formulated a problem of tracking the shifting mean of data samples. This shifting mean is a piecewise-constant step function involving two steps (seen as in black in Fig. 2 (a)), and we simulated noisy data points centered around the mean. We then used VBS with beam size $K = 2$ to fit the same latent mean model that generated the data. The color indicates the ranking among both hypotheses at each moment in time (blue as "more likely"

---

[7]This also applies to the baselines "Bayesian Forgetting" (BF) and Variational Continual Learning" (VCL) introduced in Section 4.2.

vs. orange as "less likely"). While hypothesis 1 assumes a single distribution shift (initially blue), hypothesis 2 (initially orange) assumes two shifts. We see that hypothesis 1 is initially more likely, but gets over-ruled by the better hypothesis 2 later (note the color swap at step 23).

## 4.2 Baselines

In our supervised experiments (Section 4.3 and Section 4.4), we compared VBS against adaptive methods, Bayesian online learning baselines, and independent batch learning baselines.[8] Among the adaptive methods, we formulated a supervised learning version of Bayesian online change-point detection (BOCD) [Adams and MacKay, 2007].[9] We also implemented Bayesian forgetting (BF) [Kurle et al., 2020] with convolutional neural networks for proper comparisons. Bayesian online learning baselines include variational continual learning (VCL) [Nguyen et al., 2018] and Laplace propagation (LP) [Smola et al., 2003, Nguyen et al., 2018]. Finally, we also adopt a trivial baseline of learning independent regressors/classifiers on each batch in both a Bayesian and non-Bayesian fashion. For VBS and BOCD we always report the most dominant hypothesis. In unsupervised learning experiments, we compared against the online version of word2vec [Mikolov et al., 2013] with a diffusion prior, dynamic word embeddings [Bamler and Mandt, 2017].

## 4.3 Bayesian Linear Regression Experiments

As a simple first version of VBS, we tested an online linear regression setup for which the posterior can be computed analytically. The analytical solution removes the approximation error of the variational inference procedure as well as optimization-related artifacts since closed-form updates are available. Detailed derivations are in Supplement D.

**Real Datasets with Concept Shifts.**   We investigated three real-world datasets with *concept shifts*:

- **Malware** This dataset is a collection of 100K malignous and benign computer programs, collected over 44 months [Huynh et al., 2017]. Each program has 482 counting features and a real-valued probability $p \in [0, 1]$ of being malware. We linearly predicted the log-odds.
- **SensorDrift** A collection of chemical sensor readings [Vergara et al., 2012]. We predicted the concentration level of gas *acetaldehyde*, whose 2,926 samples and 128 features span 36 months.
- **Elec2** The dataset contains the electricity price over three years of two Australian states [Harries and Wales, 1999]. While the original problem formulation used a majority vote to generate 0-1 binary labels on whether the price increases or not, we averaged the votes out into real-valued probabilities and predicted the log-odds instead. We had 45,263 samples and 14 features.

At each step, only one data sample is revealed to the regressor. We evaluated all methods with one-step-ahead absolute error[10] and computed the mean cumulative absolute error (MCAE) at every step. In Table 1, we didn't report the variance of MCAEs since there is no stochastic optimization noise. Table 1 shows that VBS has the best average of MCAEs among all methods. We also reported the running performance in Supplement G.2, where other experimental details are available as well.

**Basketball Player Tracking.**   We explored a collection of basketball player movement trajectories.[11] Each trajectory has wide variations in player velocities. We treated the trajectories as time series and used a Bayesian transition matrix to predict the next position $\mathbf{x}_{t+1}$ based on the current position $\mathbf{x}_t$. This matrix is learned and adapted on the fly for each trajectory.

We first investigated the effect of the temperature parameter $\beta$ in our approach. To this end, we visualized the detected change points on an example trajectory. We used VBS (K=1, greedy search) and compared different values of $\beta$ in Fig. 2 (b). The figure shows that the larger $\beta$, the more change

---

[8]As a reminder, a "batch" at discrete time $t$ is the dataset available for learning; on the other hand, a "mini-batch" is a small set of data used for computing gradients for stochastic gradient-based optimization.

[9]1) Although BOCD is mostly applied for unsupervised learning, its application in supervised learning and its underlying model's adaptation to change points are seldom investigated. 2) When the model is non-conjugate, such as Bayesian neural networks, we approximate the log evidence $\log p(y|x)$ by the evidence lower bound.

[10]We measured the error in the probability space for classification problems (Malware and Elec2) and the error in the data space for regression problems (SensorDrift).

[11]`https://github.com/linouk23/NBA-Player-Movements`

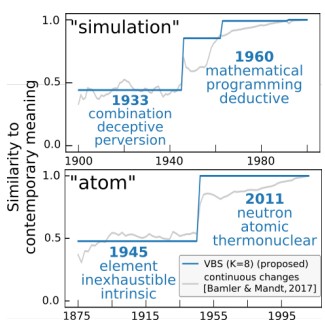

Table 1: Evaluation of Different Datasets

| Models | CIFAR-10 (Accuracy)↑ | SVHN | Malware | SensorDrift (MCAE $10^{-2}$)↓ | Elec2 | NBAPlayer (LogLike $10^{-2}$)↑ |
|---|---|---|---|---|---|---|
| VBS (K=6)[*] | **69.2±0.9** | **89.6±0.5** | **11.61** | **10.53** | 7.28 | **29.49±3.12** |
| VBS (K=3)[*] | 68.9±0.9 | 89.1±0.5 | 11.65 | 10.71 | 7.28 | 29.22±2.63 |
| VBS (K=1)[*] | 68.2±0.8 | 88.9±0.5 | 11.65 | 10.86 | **7.27** | 29.25±2.59 |
| BOCD (K=6)[♯] | 65.6±0.8 | 88.2±0.5 | 12.93 | 24.34 | 12.49 | 22.96±7.42 |
| BOCD (K=3)[♯] | 67.3±0.8 | 88.8±0.5 | 12.74 | 24.31 | 12.49 | 20.93±7.83 |
| BF[¶] | **69.8±0.8** | **89.9±0.5** | 11.71 | 11.40 | 13.37 | 24.17±2.29 |
| VCL[†] | 66.7±0.8 | 88.7±0.5 | 13.27 | 24.90 | 16.59 | 3.48±25.53 |
| LP[‡] | 62.6±1.0 | 82.8±0.9 | 13.27 | 24.90 | 16.59 | 3.48±25.53 |
| IB[§] | 63.7±0.5 | 85.5±0.7 | 16.6 | 27.71 | 12.48 | -44.87±16.88 |
| IB[§] (Bayes) | 64.5±0.3 | 87.8±0.1 | 16.6 | 27.71 | 12.48 | -44.87±16.88 |

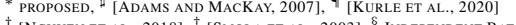

[*] PROPOSED, [♯] [ADAMS AND MACKAY, 2007], [¶] [KURLE ET AL., 2020]
[†] [NGUYEN ET AL., 2018], [‡] [SMOLA ET AL., 2003], [§] INDEPENDENT BATCH

Figure 3: Sparse word meaning changes in "simulation" and "atom".

points are detected; the smaller $\beta$, the detected change points get sparser, i.e., $\beta$ determines the model's sensitivity to changes. This observation confirms the assumption that $\beta$ controls the assumed strength of distribution shifts.

In addition, the result also implies the robustness of poorly selected $\beta$s. When facing an abrupt change in the trajectory, the regressor has two adapt options based on different $\beta$s – make a single strong adaptation or make a sequence of weak adaptations – in either case, the model ends up adapting itself to the new trajectory. In other words, people can choose different $\beta$ for a specific environment, with a trade-off between adaptation speed and the erased amount of information.

Finally, regarding the quantitative results, we evaluated all methods with the time-averaged predictive log-likelihood on a reserved test set in Table 1. Our proposed methods yield better performance than the baselines. In Supplement F, we provide more results of change point detection.

### 4.4 Bayesian Deep Learning Experiments

Our larger-scale experiments involve Bayesian convolutional neural networks trained on sequential batches for image classification using CIFAR-10 [Krizhevsky et al., 2009] and SVHN [Netzer et al., 2011]. Every few batches, we manually introduce *covariate shifts* through transforming all images globally by combining rotations, shifts, and scalings. Each transformation is generated from a fixed, predefined distribution (see Supplement G.3). The experiment involved 100 batches in sequence, where each batch contained a third of the transformed datasets. We set the temperature $\beta = 2/3$ and set the CELBO temperature $T = 20,000$ (in Eq. 6) for all supervised experiments.

Table 1 shows the performances of all considered methods and both data sets, averaged across all of the 100 batches. Within their confidence bounds, VBS and BF have comparable performances and outperform the other baselines. We conjecture that the strong performance of BF can be attributed to the fact that our imposed changes are still relatively evenly spaced and regular. The benefit of beam search in VBS is evident, with larger beam sizes consistently performing better.

### 4.5 Unsupervised Experiments

Our final experiment focused on unsupervised learning. We intended to demonstrate that VBS helps uncover interpretable latent structure in high-dimensional time series by detecting change points. We also showed that the detected change points help reduce the storage size and maintain salient features.

Towards this end, we analyzed the semantic changes of individual words over time in an unsupervised setup. We used Dynamic Word Embeddings (DWE) [Bamler and Mandt, 2017] as our base model. The model is an online version of Word2Vec [Mikolov et al., 2013]. Word2Vec projects a vocabulary into an embedding space and measures word similarities by cosine distance in that space. DWE further imposes a time-series prior on the embeddings and tracks them over time. For our proposed approach, we augmented DWE with VBS, allowing us to detect the changes of words meaning.

We analyzed three large time-stamped text corpora, all of which are available online. Our first dataset is the Google Books corpus [Michel et al., 2011] in $n$-grams form. We focused on 1900 to 2000

with sub-sampled 250M to 300M tokens per year. Second, we used the Congressional Records dataset [Gentzkow et al., 2018], which has 13M to 52M tokens per two-year period from 1875 to 2011. Third, we used the UN General Debates corpus [Jankin Mikhaylov et al., 2017], which has about 250k to 450k tokens per year from 1970 to 2018.

Our first experiments demonstrate VBS provides more interpretable step-wise word meaning shifts than the continuous shifts (DWE). Due to page limits, in Fig. 3 we selected two example words and their three nearest neighbors in the embedding space at different years. The evolving nearest neighbors reflect a semantic change of the words. We plotted the most likely hypothesis of VBS in blue and the continuous-path baseline (DWE) in grey. While people can roughly tell the change points from the continuous path, the changes are surrounded by noisy perturbations and sometimes submerged within the noise. VBS, on the other hand, makes clear decisions and outputs explicit change points. As a result, VBS discovers that the word "atom" changes its meaning from "element" to "nuclear" in 1945–the year when two nuclear bombs were detonated; word "simulation" changes its dominant context from "deception" to "programming" with the advent of computers in the 1950s. Besides interpretable changes points, VBS provides multiple plausible hypotheses (Supplement G.4).

Our second experiments exemplify the usefulness of the detected *sparse* change points, which lead to sparse segments of embeddings. The usefulness comes in two folds: 1) while alleviating the burden of the disk storage by storing one value for each segment, 2) the sparsity preserves the salient features of the original model. To illustrate these two aspects, we design a document dating task that exploits the probabilistic interpretation of word embeddings. The idea is to assign a test document to the year whose embeddings provide the highest likelihood. In Figure 2 (c), we measure the model sparsity on the x-axis with the average updated embeddings per step (The maximum is 10000, which is the vocabulary size). The feature preservation ability is measured by document dating accuracy on the y-axis. We adjust the prior log-odds $\xi_0$ (Eq. 6) to have successive models with different change point sparsity and then measure the dating accuracy. We also designed an oracle baseline named "binning" (grey, Supplement G.4). For VBS, we show the dominant hypothesis (blue) as well as the subleading hypotheses (orange). The most likely hypothesis of VBS outperforms the baseline, leading to higher document dating precision at much smaller disk storage.

## 5    Discussion

**Beyond Gaussian Posterior Approximations.**    While the Gaussian approximation is simple and is widely used (and broadly effective) in practice in Bayesian inference [e.g., Murphy [2012], pp.649-662], our formulation does not rule out the extensions to exponential families. $\tau_t$ in Eq. 2 could be generalized by reading off sufficient statistics of the previous approximate posterior. To this end, we need a sufficient statistic that is associated with some measure of entropy or variance that we broaden after each detected change. For example, the Gamma distribution can broaden its scale, and for the categorical distribution, we can increase its entropy/temperature. More intricate (e.g. multimodal) possible alternatives for posterior approximation are also possible, for example, Gaussian mixtures.

## 6    Conclusions

We introduced variational beam search: an approximate inference algorithm for Bayesian online learning on non-stationary data with irregular changes. Our approach mediates the tradeoff between a model's ability to memorize past data while still being able to adapt to change. It is based on a Bayesian treatment of a given model's parameters and aimed at tuning them towards the most recent data batch while exploiting prior knowledge from previous batches. To this end, we introduced a sequence of a discrete change variables whose value controlled the way we regularized the model. For no detected change, we regularized the new learning task towards the previously learned solution; for a detected change, we broadened the prior to give room for new data evidence. This procedure is combined with beam search over the discrete change variables. In different experiments, we showed that our proposed model (1) achieved lower error in supervised setups, and (2) revealed a more interpretable and compressible latent structure in unsupervised experiments.

**Broader Impacts.**    As with many machine learning algorithms, there is a danger that more automation could potentially result in unemployment. Yet, more autonomous adaptation to changes will enhance the safety and robustness of deployed machine learning systems, such as self-driving cars.

# Acknowledgements

We gratefully acknowledge extensive contributions from Robert Bamler (previously UCI, now at the University of Tübingen), which were indispensable to this work.

This material is based upon work supported by the National Science Foundation under the CAREER award 2047418 and grant numbers 1633631, 1928718, 2003237, and 2007719; by the National Science Foundation Graduate Research Fellowship under grant number DGE-1839285; by the Defense Advanced Research Projects Agency (DARPA) under Contract No. HR001120C0021; by an Intel grant; and by grants from Qualcomm. Any opinion, findings, and conclusions or recommendations expressed in this material are those of the authors and do not necessarily reflect the views of the National Science Foundation, nor do they reflect the views of DARPA. Additional revenues potentially related to this work include: research funding from NSF, NIH, NIST, PCORI, and SAP; fellowship funding from HPI; consulting income from Amazon.com.

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
