# Detecting and Adapting to Irregular Distribution Shifts in Bayesian Online Learning: Supplementary Materials

## A    Structured Variational Inference

According to the main paper, we consider the generative model $p(\mathbf{x}_t, \mathbf{z}_t, s_t | \mathbf{x}_{1:t-1}, s_{1:t-1}) = p(s_t)p(\mathbf{z}_t|s_t; \boldsymbol{\tau}_t)p(\mathbf{x}_t|\mathbf{z}_t)$ at time step $t$, where the dependence on $\mathbf{x}_{1:t-1}, s_{1:t-1}$ is contained in $\boldsymbol{\tau}_t$. Upon observing the data $\mathbf{x}_t$, both $\mathbf{z}_t$ and $s_t$ are inferred. However, exact inference is not available due to the intractability of the marginal likelihood $p(\mathbf{x}_t|s_{1:t}, \mathbf{x}_{1:t-1})$. To tackle this, we utilize structured variational inference for both the latent variables $\mathbf{z}_t$ and the Bernoulli change variable $s_t$. To this end, we define the joint variational distribution $q(\mathbf{z}_t, s_t) = q(s_t|s_{1:t-1})q(\mathbf{z}_t|s_{1:t})$ as in the main paper. For notational simplicity, we omit the dependence on $s_{1:t-1}$. Then the updating procedure for $q(s_t)$ and $q(\mathbf{z}_t|s_t)$ is obtained by maximizing the ELBO $\mathcal{L}(q)$:

$$q^*(\mathbf{z}_t, s_t) = \underset{q(\mathbf{z}_t, s_t) \in Q}{\arg\max} \ \mathcal{L}(q),$$

$$\mathcal{L}(q) := \mathbb{E}_q[\log p(\mathbf{x}_t, \mathbf{z}_t, s_t; \boldsymbol{\tau}_t) - \log q(\mathbf{z}_t, s_t)].$$

Given the generative models, we can further expand $\mathcal{L}(q)$ to simplify the optimization:

$$\begin{aligned}
\mathcal{L}(q) &= \mathbb{E}_{q(s_t)q(\mathbf{z}_t|s_t)}[\log p(s_t) + \log p(\mathbf{z}_t|s_t; \boldsymbol{\tau}_t) + \log p(\mathbf{x}_t|\mathbf{z}_t) - \log q(s_t) - \log q(\mathbf{z}_t|s_t)] \\
&= \mathbb{E}_{q(s_t)}[\log p(s_t) - \log q(s_t) + \mathbb{E}_{q(\mathbf{z}_t|s_t)}[\log p(\mathbf{z}_t|s_t; \boldsymbol{\tau}_t) + \log p(\mathbf{x}_t|\mathbf{z}_t) - \log q(\mathbf{z}_t|s_t)]] \\
&= \mathbb{E}_{q(s_t)}[\log p(s_t) - \log q(s_t) + \mathbb{E}_{q(\mathbf{z}_t|s_t)}[\log p(\mathbf{x}_t|\mathbf{z}_t)] - \mathrm{KL}(q(\mathbf{z}_t|s_t)||p(\mathbf{z}_t|s_t; \boldsymbol{\tau}_t))] \\
&= \mathbb{E}_{q(s_t)}[\log p(s_t) - \log q(s_t) + \mathcal{L}(q|s_t)]
\end{aligned} \tag{1}$$

where the second step pushes inside the expectation with respect to $q(\mathbf{z}_t|s_t)$, the third step re-orders the terms, and the final step utilizes the definition of CELBO (Eq. 7 in the main paper).

Maximizing Eq. 1 therefore implies a two-step optimization: first maximize the CELBO $\mathcal{L}(q|s_t)$ to find the optimal $q^*(\mathbf{z}_t|s_t = 1)$ and $q^*(\mathbf{z}_t|s_t = 0)$, then compute the Bernoulli distribution $q^*(s_t)$ by maximizing $\mathcal{L}(q)$ while the CELBOs $\mathcal{L}(q^*|s_t)$ are fixed.

While $q^*(\mathbf{z}_t|s_t)$ typically needs to be inferred by black box variational inference [Ranganath et al., 2014, Kingma and Welling, 2014, Zhang et al., 2018], the optimal $q^*(s_t)$ has a closed-form solution and bears resemblance to the exact inference counterpart (Eq. 4 in the main paper). To see this, we assume $\mathcal{L}(q^*|s_t)$ are given and $q(s_t)$ is parameterized by $m \in \mathbb{R}$ (for the Bernoulli distribution). Rewriting Eq. 1 gives

$$\begin{aligned}
\mathcal{L}(q) &= m(\log p(s_t = 1) - \log m + \mathcal{L}(q^*|s_t = 1)) \\
&\quad + (1 - m)(\log p(s_t = 0) - \log(1 - m) + \mathcal{L}(q^*|s_t = 0))
\end{aligned}$$

which is concave since the second derivative is negative. Thus taking the derivative and setting it to zero leads to the optimal solution of

$$\log \frac{m}{1 - m} = \log p(s_t = 1) - \log p(s_t = 0) + \mathcal{L}(q^*|s_t = 1)) - \mathcal{L}(q^*|s_t = 0)),$$

$$m = \sigma(\mathcal{L}(q^*|s_t = 1)) - \mathcal{L}(q^*|s_t = 0)) + \xi_0),$$

which attains the closed-form solution as stated in Eq. 6 in the main paper without temperature $T$.

35th Conference on Neural Information Processing Systems (NeurIPS 2021).

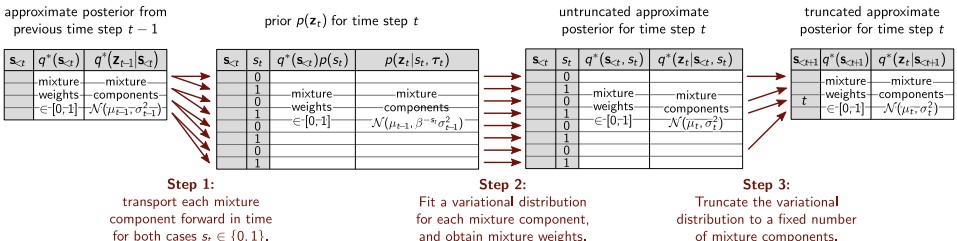

Figure 1: Conditional probability table of variational beam search

---

**Algorithm 1** Variational Beam Search

---

**Input:** task set $\{\mathbf{x}_t\}_1^T$; beam size $K$; prior log-odds $\xi_0$; conditional ELBO temperature $T$
**Output:** approximate posterior distributions $\{q^*(s_{1:t}), q^*(\mathbf{z}_t|s_{1:t})\}_1^T$
1: $q^*(\mathbf{z}_1) = \arg\max \mathbb{E}_q[\log p(\mathbf{x}_1|\mathbf{z}_1)] - \mathrm{KL}(q(\mathbf{z}_1)||p(\mathbf{z}_1))$;
2: $q^*(s_1 = 0) := 1$; $q^*(\mathbf{z}_1|s_1) := q^*(\mathbf{z}_1)$; $\mathbb{B} = \{s_1 = 0\}$;
3: **for** $t = 1, \cdots, T$ **do**
4:     $\mathbb{B}' = \{\}$
5:     **for** each hypothesis $\mathbf{s}_{<t} \in \mathbb{B}$ **do**
6:         $p(s_t = 1) := \sigma(\xi_0)$ for random variable $s_t \in \{0, 1\}$
7:         $\mathbb{B}' := \mathbb{B}' \cup \{(\mathbf{s}_{<t}, s_t = 0), (\mathbf{s}_{<t}, s_t = 1)\}$;
8:         compute the task $t$'s prior $p(\mathbf{z}_t|s_t, \boldsymbol{\tau}_t)$ (Eq. 3);
9:         perform structured variational inference (Eq. 7 and Eq. 6) given observation $\mathbf{x}_t$, resulting in $q^*(s_t, \mathbf{z}_t|\mathbf{s}_{<t}) = q^*(s_t|\mathbf{s}_{<t})q^*(\mathbf{z}_t|\mathbf{s}_{<t+1})$ where $q^*(\mathbf{z}_t|\mathbf{s}_{<t+1}$ is stored as output $q^*(\mathbf{z}_t|s_{1:t})$;
10:       approximate new hypotheses' posterior probability $p(s_{1:t}|\mathbf{x}_{1:t}) \approx q^*(\mathbf{s}_{<t}, s_t) = q^*(\mathbf{s}_{<t})q^*(s_t|\mathbf{s}_{<t})$;
11:     **end for**
12:     $\mathbb{B} := \mathtt{diverse\_truncation}(\mathbb{B}', q^*(\mathbf{s}_{<t}, s_t))$;
13:     normalize $q^*(\mathbf{s}_{<t}, s_t)$ where $(\mathbf{s}_{<t}, s_t) \in \mathbb{B}$;
14: **end for**

---

# B   Additive vs. Multiplicative Broadening

There are several possible choices for defining an informative prior corresponding to $s_t = 1$. In latent time series models, such as Kalman filters [Kalman, 1960, Bamler and Mandt, 2017], it is common to define a linear transition model $\mathbf{z}_t = A\mathbf{z}_{t-1} + \boldsymbol{\epsilon}_t$ where $\mathbf{z}_{t-1} \sim \mathcal{N}(\mathbf{z}_{t-1}; \boldsymbol{\mu}_{t-1}, \Sigma_{t-1})$ and $\boldsymbol{\epsilon}_t \sim \mathcal{N}(\boldsymbol{\epsilon}; \mathbf{0}, \Sigma_n)$. Propagating the posterior at time $t-1$ to the prior at time $t$ results in $\mathbf{z}_t \sim \mathcal{N}(\mathbf{z}_t; A\boldsymbol{\mu}_{t-1}, A\Sigma_{t-1}A^\top + \Sigma_n)$. To simplify the discussion, we set $A = I$ and $\Sigma_n = \sigma_n^2 I$; the same argument also applies for the more general case. Adding a constant noise $\boldsymbol{\epsilon}_t$ results in adding the variance of all variables with a constant $\sigma_n^2$. We thus call this convolution scheme *additive broadening*. The problem with such a choice, however, is that the associated information loss is not homogeneously distributed: $\sigma_n^2$ ignores the uncertainty in $\mathbf{z}_t$, and dimensions of $\mathbf{z}_t$ with low posterior uncertainty lose more information relative to dimensions of $\mathbf{z}_t$ that are already uncertain. We found that this scheme deteriorates the learning signal.

We therefore consider *multiplicative broadening* (or *relative broadening* since the associated information loss depends on the original variance) as *tempering* described in the main paper, resulting in $p(\mathbf{z}_t|s_t, \boldsymbol{\tau}_t) \propto p(\mathbf{z}_{t-1}|\mathbf{x}_{1:t-1}, s_{1:t-1})^\beta$ for $\beta > 0$. For a Gaussian distribution, the resulting variance scales the original variance with $\frac{1}{\beta}$. In practice, we found relative or multiplicative broadening to perform much better and robustly than additive posterior broadening. Since tempering broadens the posterior non-locally, this scheme does not possess a continuous latent time series interpretation [1].

# C   Details on "Shy" Variational Greedy Search and Variational Beam Search

---

[1]This means that it is impossible to specify a conditional distribution $p(\mathbf{z}_t|\mathbf{z}_{t-1})$ that corresponds to relative broadening.

**"Shy" Variational Greedy Search.** As illustrated in Fig. 1 in the main text, one obtains better interpretation if one outputs the variational parameters $\mu_t$ and $\sigma_t$ at the end of a segment of constant $\mathbf{z}_t$. More precisely, when the algorithm detects a change point $s_t = 1$, it outputs the variational parameters $\mu_{t-1}$ and $\sigma_{t-1}$ from just before the detected change point $t$. These parameters define a variational distribution that has been fitted, in an iterative way, to all data points since the preceding detected change point. We call this the "shy" variant of the variational greedy search algorithm, because this variant quietly iterates over the data and only outputs a new fit when it is as certain about it as it will ever be. The red lines and regions in Fig. 1 (a) in the main paper illustrate means and standard deviations outputted by the "shy" variant of variational greedy search.

We applied this "Shy" variant to our illustrative example (Section 4.1) and unsupervised learning experiments (Section 4.5).

**Variational Beam Search.** As follows, we present a more detailed explanation of the variational beam search procedure mentioned in Section 3.3 of the main paper. Our beam search procedure defines an effective way to search for potential hypotheses with regards to sequences of inferred change points. The procedure is completely defined by detailing three sequential steps, that when executed, take a set of hypotheses found at time step $t - 1$ and transform them into the resulting set of likely hypotheses for time step $t$ that have appropriately accounted for the new data seen at $t$. The red arrows in Figure 1 illustrate these three steps for beam search with a beam size of $K = 4$.

In Figure 1, each of the three steps maps a table of considered histories to a new table. Each table defines a mixture of Gaussian distributions where each mixture component corresponds to a different history and is represented by a different row in the table. We start on the left with the (truncated) variational distribution $q^*(\mathbf{z}_{t-1})$ from the previous time step, which is a mixture over $K = 4$ Gaussian distributions. Each mixture component (row in the table) is labeled by a 0-1 vector $\mathbf{s}_{<t} \equiv (s_1, \cdots, s_{t-1})$ of the change variable values according to that history. Each mixture component $\mathbf{s}_{<t}$ further has a mixture weight $q^*(\mathbf{s}_{<t}) \in [0, 1]$, a mean, and a standard deviation.

We then obtain a prior for time step $t$ by transporting each mixture component of $q^*(\mathbf{z}_{t-1})$ forward in time via the broadening functional ("Step 1" in the above figure). The prior $p(\mathbf{z}_t)$ (second table in the figure) is a mixture of $2K$ Gaussian distributions because each previous history splits into two new ones for the two potential cases $s_t \in \{0, 1\}$. The label for each mixture component (table row) is a new vector $(\mathbf{s}_{<t}, s_t)$ or $\mathbf{s}_{<t+1}$, appending $s_t$ to the tail of $\mathbf{s}_{<t}$.

"Step 2" in the above figure takes the data $\mathbf{x}_t$ and fits a variational distribution $q^*(\mathbf{z}_t)$ that is also a mixture of $2K$ Gaussian distributions. To learn the variational distribution, we (i) numerically fit each mixture component $q(\mathbf{z}_t|\mathbf{s}_{<t}, s_t)$ individually, using the corresponding mixture component of $p(\mathbf{z}_t)$ as the prior; (ii) evaluate (or estimate) the CELBO of each fitted mixture component, conditioned on $(\mathbf{s}_{<t}, s_t)$; (iii) compute the approximate posterior probability $q^*(s_t|\mathbf{s}_{<t})$ of each mixture component, in the presence of the CELBOs; and (iv) obtain the mixture weight equal to the posterior probability over $(\mathbf{s}_{<t}, s_t)$, i.e., $p(s_{1:t}|\mathbf{x}_{1:t})$, best approximated by $q^*(\mathbf{s}_{<t})q^*(s_t|\mathbf{s}_{<t})$.

"Step 3" in the above figure truncates the variational distribution by discarding $K$ of the $2K$ mixture components. The truncation scheme can be either the "vanilla" beam search or diversified beam search outlined in the main paper. The truncated variational distribution $q_t(\mathbf{z}_t)$ is again a mixture of only $K$ Gaussian distributions, and it can thus be used for subsequent update steps, i.e., from $t$ to $t + 1$.

The pseudocode is listed in Algo 1.

# D  Online Bayesian Linear Regression with Variational Beam Search

This section will derive the analytical solution of online updates for both Bayesian linear regression and the probability of change points. We consider Gaussian prior distributions for weights. The online update of the posterior distribution is straightforward in the natural parameter space, where the update is analytic given the sufficient statistics of the observations. If we further allow to temper the weights' distributions with a fixed temperature $\beta$, then this corresponds to multiplying each element in the precision matrix by $\beta$. We applied this algorithm to the linear regression experiments in Section 4.3. For unified names, we still use the word "variational" even though the solutions are analytical.

## D.1 Variational Continual Learning for Online Linear Regression

Let's start with assuming a generative model at time $t$:

$$\boldsymbol{\theta} \sim \mathcal{N}(\boldsymbol{\mu}, \Sigma),$$
$$y_t = \boldsymbol{\theta}^\top \mathbf{x}_t + \epsilon, \quad \epsilon \sim \mathcal{N}(0, \sigma_n^2), \tag{2}$$

and the noise $\epsilon$ is constant over time.

The posterior distribution of $\boldsymbol{\theta}$ is of interest, which is Gaussian distributed since both the likelihood and prior are Gaussian. To get an online recursion for $\boldsymbol{\theta}$'s posterior distribution over time, we consider the natural parameterization. The prior distribution under this parameterization is

$$
\begin{aligned}
p(\boldsymbol{\theta}) &= \frac{1}{Z} \exp\left( -\frac{(\boldsymbol{\theta} - \boldsymbol{\mu})^\top \Sigma^{-1}(\boldsymbol{\theta} - \boldsymbol{\mu})}{2} \right) \\
&= \frac{1}{Z} \exp\left( -\frac{1}{2}\boldsymbol{\theta}^\top \Sigma^{-1}\boldsymbol{\theta} + \boldsymbol{\theta}^\top \Sigma^{-1}\boldsymbol{\mu} \right) \\
&= \frac{1}{Z} \exp\left( -\frac{1}{2}\boldsymbol{\theta}^\top \Lambda\boldsymbol{\theta} + \boldsymbol{\theta}^\top \boldsymbol{\eta} \right)
\end{aligned}
$$

where $\Lambda = \Sigma^{-1}, \boldsymbol{\eta} = \Sigma^{-1}\boldsymbol{\mu}$ are the natural parameters and the terms unrelated to $\boldsymbol{\theta}$ are absorbed into the normalizer $Z$.

Following the same parameterization, the posterior distribution can be written

$$
\begin{aligned}
p(\boldsymbol{\theta}|\mathbf{x}_t, y_t) &\propto p(\boldsymbol{\theta})p(y_t|\mathbf{x}_t, \boldsymbol{\theta}) \\
&= \frac{1}{Z} \exp\left( -\frac{1}{2}\boldsymbol{\theta}^\top \Lambda\boldsymbol{\theta} + \boldsymbol{\theta}^\top \boldsymbol{\eta} - \frac{1}{2}\sigma_n^{-2}\boldsymbol{\theta}^\top (\mathbf{x}_t\mathbf{x}_t^\top)\boldsymbol{\theta} + \sigma_n^{-2}y_t\boldsymbol{\theta}^\top \mathbf{x}_t \right) \\
&= \frac{1}{Z} \exp\left( -\frac{1}{2}\boldsymbol{\theta}^\top (\Lambda + \sigma_n^{-2}\mathbf{x}_t\mathbf{x}_t^\top)\boldsymbol{\theta} + \boldsymbol{\theta}^\top (\boldsymbol{\eta} + \sigma_n^{-2}y_t\mathbf{x}_t) \right).
\end{aligned}
$$

Thus we get the recursion over the natural parameters

$$
\begin{aligned}
\Lambda' &= \Lambda + \sigma_n^{-2}\mathbf{x}_t\mathbf{x}_t^\top, \\
\boldsymbol{\eta}' &= \boldsymbol{\eta} + \sigma_n^{-2}y_t\mathbf{x}_t,
\end{aligned}
$$

from which the posterior mean and covariance can be solved.

## D.2 Prediction and Marginal Likelihood

We can get the posterior predictive distribution for a new input $\mathbf{x}_*$ through inspecting Eq. 2 and utilizing the linear properties of Gaussian. Assuming the generative model as specified above, we replace $\mathbf{x}_t$ with $\mathbf{x}_*$ in Eq. 2. Since $\boldsymbol{\theta}$ is Gaussian distributed, by its linear property, $\mathbf{x}_*^\top \boldsymbol{\theta}$ conforms to $\mathcal{N}(\mathbf{x}_*^\top \boldsymbol{\theta}; \mathbf{x}_*^\top \boldsymbol{\mu}, \mathbf{x}_*^\top \Sigma \mathbf{x}_*)$. Then the addition of two independent Gaussian results in $y_* \sim \mathcal{N}(y_*; \mathbf{x}_*^\top \boldsymbol{\mu}, \sigma_n^2 + \mathbf{x}_*^\top \Sigma \mathbf{x}_*)$.

The marginal likelihood shares this same form with the posterior predictive distribution, with a potentially different pair of sample $(\mathbf{x}, y)$. To see this, given a prior distribution $\boldsymbol{\theta} \sim \mathcal{N}(\boldsymbol{\theta}; \boldsymbol{\mu}, \Sigma)$, then the marginal likelihood of $y|\mathbf{x}$ is

$$
\begin{aligned}
p(y|\mathbf{x}; \boldsymbol{\mu}, \Sigma, \sigma_n) &= \int p(y|\mathbf{x}, \boldsymbol{\theta})p(\boldsymbol{\theta}; \boldsymbol{\mu}, \Sigma)d\boldsymbol{\theta} \\
&= \mathcal{N}(y; \mathbf{x}^\top \boldsymbol{\mu}, \sigma_n^2 + \mathbf{x}^\top \Sigma \mathbf{x})
\end{aligned}
\tag{3}
$$

with $\sigma_n^2$ being the noise variance. Note that in variational inference with an intractable marginal likelihood (not like the linear regression here), this is the approximated objective (Evidence Lower Bound (ELBO), indeed) we aim to maximize.

**Computation of the Covariance Matrix** Since we parameterize the precision matrix instead of the covariance matrix, the variance of the new test sample requires to take the inverse of the precision matrix. In order to do this, we employ the eigendecomposition of the precision matrix and re-assemble to the covariance matrix through inverting the eigenvalues. A better approach is to apply the Sherman-Morrison formula for the rank one update[2], which can reduce the computation from $O(n^3)$ to $O(n^2)$.

**Logistic Normal Model** If we are modeling the log-odds by a Bayesian linear regression, then we need to map the log-odds to the interval $[0, 1]$ by the sigmoid function, to make it a valid probability. Specifically, suppose $a = \mathcal{N}(a; \mu_a, \sigma_a^2)$ and $y = \sigma(a)$ (note we abuse $\sigma$ by variances and functions, but it is clear from the context and the subscripts) where $\sigma(\cdot)$ is a logistic sigmoid function. Then $y$ has a logistic normal distribution. Given $p(y)$, we can make decisions for the value of $y$. There are three details that worth noting. First is from the non-linear mapping of $\sigma(\cdot)$. One special property of $p(y)$ is that $p(y)$ can be bimodal if the base variance or $\sigma_a$ is large. A consequence is that the mode of $p(a)$ does not necessarily correspond to $p(y)$'s mode and $\mathbb{E}[y] \neq \sigma(\mathbb{E}[a])$. Second is for the binary classification: the decision boundary of $y$, i.e., 0.5, is consistent with the one of $x$, i.e., 0, for decisions either by $\mathbb{E}[y]$ or by $\mathbb{E}[a]$. See Rusmassen and Williams [2005] (Section 3.4) and Bishop et al. [1995] (Section 10.3). Third, if our loss function for decision making is the absolute error, then the best prediction is the median of $y = \sigma(\hat{a})$ [Friedman et al., 2001] where $\hat{a}$ is the median of $a$. This follows from the monotonicity of $\sigma(\cdot)$ that does not change the order statistics.

### D.3 Inference over the Change Variable

To infer the posterior distribution of $s_t$ given observations $(\mathbf{x}_{1:t}, y_{1:t})$, we apply Bayes' theorem to infer the posterior log-odds as in the main paper

$$\log\left(\frac{p(s_t = 1|\mathbf{x}_{1:t}, y_{1:t}, s_{1:t-1})}{p(s_t = 0|\mathbf{x}_{1:t}, y_{1:t}, s_{1:t-1})}\right) = \frac{\log(p(\mathbf{x}_t|\mathbf{x}_{1:t-1}, y_{1:t}, s_t = 1, s_{1:t-1}))}{\log(p(\mathbf{x}_t|\mathbf{x}_{1:t-1}, y_{1:t}, s_t = 0), s_{1:t-1})} + \xi_0$$

where $p(\mathbf{x}_t|\mathbf{x}_{1:t-1}, y_{1:t}, s_t)$ is exactly Eq. 3 but has different parameter values dictated by $s_t$ and $\beta$.

In the next part, we show the resulting distribution of the broadening operation.

#### D.3.1 Tempering a Multivariate Gaussian

We will show the tempering operation of a multivariate Gaussian corresponds to multiplying each element in the precision matrix by the fixed temperature, a simple form in the natural space.

Suppose we allow to temper / broaden the $\boldsymbol{\theta}$'s multivariate Gaussian distribution before the next time step, accommodating more evidence. Let the broadening constant or temperature be $\beta \in (0, 1]$. We derive how $\beta$ affects the multivariate Gaussian precision.

Write the tempering explicitly,

$$p(\boldsymbol{\theta})^\beta = \frac{1}{Z} \exp\left(-\frac{1}{2}\beta(\boldsymbol{\theta} - \boldsymbol{\mu})^\top \Lambda(\boldsymbol{\theta} - \boldsymbol{\mu})\right)$$

$$= \frac{1}{Z} \exp\left(-\frac{1}{2}(\boldsymbol{\theta} - \boldsymbol{\mu})^\top \Lambda_\beta(\boldsymbol{\theta} - \boldsymbol{\mu})\right)$$

among which we are interested in the relationship between $\Lambda_\beta$ and $\Lambda$ and $\beta$. To this end, re-write the quadratic form in the summation

$$\beta(\boldsymbol{\theta} - \boldsymbol{\mu})^\top \Lambda(\boldsymbol{\theta} - \boldsymbol{\mu})$$

$$= \sum_{i,j} \beta\Lambda_{ij}(\boldsymbol{\theta}_i - \boldsymbol{\mu}_i)(\boldsymbol{\theta}_j - \boldsymbol{\mu}_j)$$

$$= \sum_{i,j} \Lambda_{\beta,ij}(\boldsymbol{\theta}_i - \boldsymbol{\mu}_i)(\boldsymbol{\theta}_j - \boldsymbol{\mu}_j)$$

where we can identify an element-wise relation: for all possible $i, j$

$$\Lambda_{\beta,ij} = \beta\Lambda_{ij}.$$

---

[2]`https://en.wikipedia.org/wiki/Sherman%E2%80%93Morrison_formula`

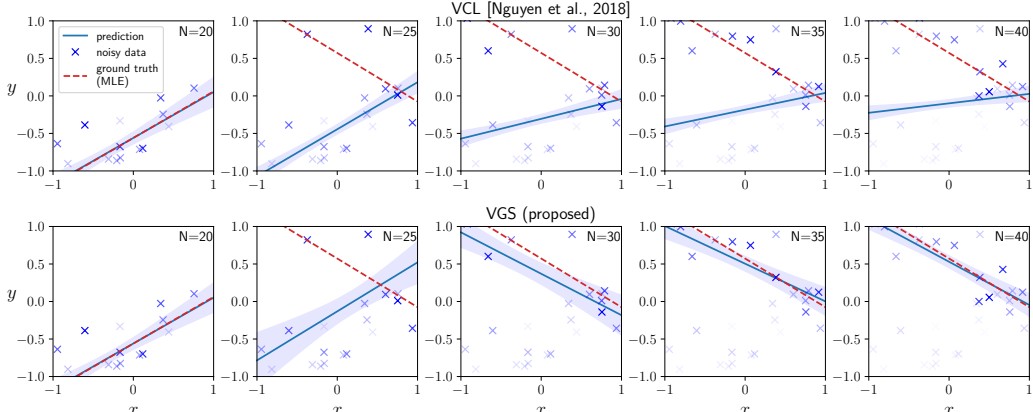

Figure 2: 1D online Bayesian linear regression with distribution shift. More recent samples are colored darker. Due to the catastrophic remembering, VCL fails to adapt to new observations.

### D.3.2 Prediction

As above, we are interested in the posterior predictive distribution for a new test sample $(\mathbf{x}_*, y_*)$ after absorbing evidence. Let's denote the parameters of $\boldsymbol{\theta}$'s posterior distributions by $\boldsymbol{\mu}_{s_{1:t}}$ and $\Sigma_{s_{1:t}}$, where the dependence over $s_{1:t}$ is made explicit. We then make posterior predictions with each component

$$p(y_*|\mathbf{x}_*, \mathbf{x}, y, s_{1:t}) = \mathcal{N}(y_*; \mathbf{x}_*^\top \boldsymbol{\mu}_{s_{1:t}}, \sigma_n^2 + \mathbf{x}_*^\top \Sigma_{s_{1:t}} \mathbf{x}_*).$$

## E Visualization of Catastrophic Remembering Effects

In order to demonstrate the effect of catastrophic remembering, we consider a simple linear regression model. We will see that, when the data distribution changes, a Bayesian online learning framework becomes quickly overconfident and unable to adjust to the changing data distribution. On the other hand, with tempering, variational greedy search (VGS) can partially forget the previous knowledge and then adapt to the shifted distribution.

**Data Generating Process** We generated the samples by the following generative model:

$$x \sim \text{Unif}(-1, 1),$$
$$y \sim \mathcal{N}(f(x), 0.1^2)$$

where $f(x)$ equals $f_1(x) = 0.7x - 0.5$ or $f_2(x) = -0.7x + 0.5$. In this experiments, we sampled the first 20 points from $f_1$ and the remaining 20 points from $f_2$.

**Model Parameters** We applied the Bayes updates mentioned in Section D to do inference over the slope and intercept. We set the initial priors of the weights to be standard Gaussian and the observation noise $\sigma_n^2$ to be the true scale, 0.1. This setting is enough for VCL.

For VGS, we set the same noise variance $\sigma_n^2 = 0.1$. For the method-specific parameters, we set $\xi_0 = \log(0.35/(1 - 0.35))$ and $\beta = 1/3.5$.

We plotted the noise-free posterior predictive distribution for both VCL and VGS. That is, let $f_*(x)$ be the fitted function, we plotted $p(f_*(x_t)|x_t, \mathcal{D}_{1:t-1}) = \int p(f_*(x_t)|x_t, \mathbf{w}, \mathcal{D}_{1:t-1})p(\mathbf{w}|\mathcal{D}_{1:t-1})d\mathbf{w}$ where $\mathcal{D}_{1:t-1}$ is the observed samples so far.

**Results** We first visualized the catastrophic remembering effect through a 1D online Bayesian linear regression task where a distribution shift occurred unknown to the regressor (Fig. 2). In this setup, noisy data were sampled from two ground truth functions $f_1(x) = 0.7x - 0.5$ and $f_2(x) = -0.7x + 0.5$, where, with constant additive noise, the first 20 samples came from $f_1$ and the remaining 20 samples were from $f_2$. The observed sample is presented one by one to the regressor.

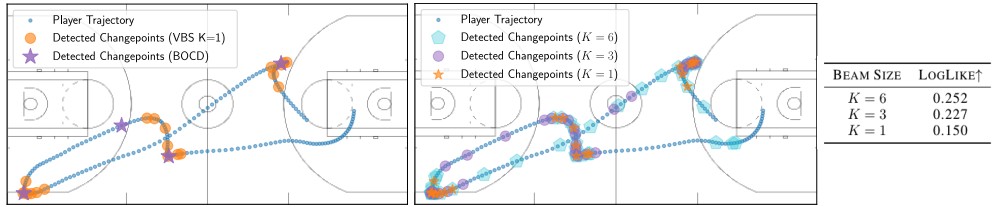

Figure 3: Changepoints in Basketball player movement tracking. (**Left plot**) Comparisons between VBS and BOCD [Adams and MacKay, 2007]. While BOCD detect change points at sparse, abrupt changes, VBS detects the changes at smooth, gradual changes. (**Right plot**) Ablation study over beam size $K$ for VBS while fixing other parameters. As we increase the beam size, qualitatively different change points are detected and the predictive likelihood improves.

Before the regression starts, the weights (slope and intercept) were initialized to be standard Gaussian distributions. We experimented two different online regression methods, original online Bayesian linear regression (VCL [Nguyen et al., 2018]) and the proposed variational greedy search (VGS). In Fig. 2, to show a practical surrogate for the ground truth, we plotted the maximum likelihood estimation (MLE) for each function given the observations. The blue line and the shaded area correspond to the mean and standard deviation of the posterior predictive distribution after observing $N$ samples. As shown in Fig. 2, both VCL (top) and VGS (bottom) faithfully fit to $f_1$ after observing the first 20 samples. However, when another 20 new observations are sampled from $f_2$, VCL shows catastrophic remembering of $f_1$ and cannot adapt to $f_2$. VGS, on the other hand, tempers the prior distribution automatically and succeeds in adaptation.

**Theoretical Explanation**    The posterior shrinkage phenomenon can be seen from the following identity

$$\mathrm{Var}(\mathbf{z}) = \mathbb{E}[\mathrm{Var}(\mathbf{z}|\mathbf{x})] + \mathrm{Var}(\mathbb{E}[\mathbf{z}|\mathbf{x}]),$$

where we denote all observations by $\mathbf{x}$ and model parameters by $\mathbf{z}$.

Both terms in the right hand side are larger than zero, thus $\mathrm{Var}(\mathbf{z}) \geq \mathbb{E}[\mathrm{Var}(\mathbf{z}|\mathbf{x})]$. This inequality says the prior variance is larger than the expected posterior variance.

It implies in a non-stationary online learning, the prior shrinks over time, and the overly strong prior delivers a misspecification for the future data.

## F    NBAPlayer: Change Point Detection Comparisons

**VBS vs. BOCD**    We investigated the changepoint detection characteristics of our proposed methods and compared against the BOCD baseline in Fig. 3 (left). On the shown example trajectory, BOCD detects abrupt change points, corresponding to different plays, a similar phenomenon observed by [Harrison et al., 2020]. However, we argue that it is insufficient and late to identify a player's strategy purpose – it only triggers an alarm after a new play starts. VBS, on the other hand, characterizes the transition phases between plays, triggers an early alarm before the next play starts. It also shows the difference between BOCD and VBS in changepoint detection: while BOCD only detects abrupt changes, VBS detects gradual changes as well.

**Practical Considerations of VBS and BOCD**    Using our variational inference extensions of BOCD, we can overcome the inference difficulty of non-conjugate models. But, considering practical issues, VBS is better in that the detected change points are easy to read from the binary change variable values $s_{1:t}$ and free from post-processing – a procedure that BOCD has to exercise. BOCD often outputs a sequence of run lengths either online or offline, among which the change points do not always correspond to the time when the most probable run length becomes one. Instead, the run length oftentimes is larger than one when change point happens. Then people have to inspect the run lengths and set a subjective threshold to determine when a change point occurs, which is not data-driven and may incur undesirable detection. For example, in our basketball player tracking experiments, we set the threshold of change points to be 50; VBS, on the other hand, is free from this post-process thresholding and provides multiple plausible, completely data-driven hypotheses of change points.

**Ablation Study of Beam Size**  The right plot and the table in Fig 3 shows, on the example trajectory, the detected change points and the average log-likelihood as the beam size $K$ changes. When $K = 1$, VBS characterizes the trajectory where the velocity direction changes; when $K = 3$ or 6, it seems that some parts where the velocity value changes are detected. We also observed that the average predictive log-likelihood improves as $K$ increases.

# G  Experiment Details and Results

In this section, we provide the unstated details of the experiments mentioned in the main paper. These details include but are not limited to hardware infrastructure used to experiment, physical running time, hyperparameter searching, data generating process, evaluation metric, additional results, empirical limitations, and so on. The subsection order corresponds to the experiments order in the main paper. We first provide some limitations of our methods during experiments.

**Limitations**  Our algorithm is theoretically sound. The generality and flexibility renders a great performance in experiments, however, at the expense of taking more time to search the hyperparameters in a relatively large space. Specifically, there are two hyperparameters to tune: $\xi_0$ and $\beta$. The grid search over these two hyperparameters could be slow. When we further take into account the beam size $K$, it adds more burden in parameter searching. But we give a reference scope to the tuning region where the search should perform. Oftentimes, we use the same parameters across beam sizes.

## G.1  An Illustrative Example

**Data Generating Process**  To generate Figure 2 in the main paper, we used a step-wise function as ground truth, where the step size was 1 and two step positions were chosen randomly. We sampled 30 equally-spaced points with time spacing 1. To get noisy observations, Gaussian noise with standard deviation 0.5 was added to the points.

**Model Parameters**  In this simple one-dimensional model, we used absolute broadening with a Gaussian transition kernel $K(\mathbf{z}_t, \mathbf{z}'_t) = \mathcal{N}(\mathbf{z}_t - \mathbf{z}'_t, D\Delta t)$ where $D = 1.0$ and $\Delta t = 1$. The inference is thus tractable because $p(\mathbf{z}_t|s_t)$ is conditional conjugate to $p(\mathbf{x}_t|\mathbf{z}_t, s_t)$ (and both are Gaussian distributed). We set the prior log-odds $\epsilon_0$ to $\log \frac{p(s_t=1)}{p(s_t=0)}$, where $p(s_t = 1) = 0.1$. We used beam size 2 to do the inference.

## G.2  Bayesian Linear Regression Experiments

We performed all linear regression experiments on a laptop with 2.6 GHz Intel Core i5 CPU. All models on SensorDrift, Elec2, and NBAPlayer dataset finished running within 5 minutes. Running time on Malware dataset varied: VCL, BF, and Independent task are within 10 minutes; VGS takes about two and half hours; VBS (K=3) takes about six hours; VBS (K=6) takes about 12 hours. BOCD takes similar time with VBS. The main difference between VCL's computation cost and VBS's computation cost lies in the necessarity of inverting the precision matrix into covariance matrix. However, the matrix inverse computation in VBS and BOCD can be substantially reduced from $O(n^3)$ to $O(n^2)$ by the recursion of Sherman–Morrison formula; we will implement this in the future.

**Problem Definitions**  We considered both classification experiments (Malware, Elec2) and regression experiments (SensorDrift, NBAPlayer). The classification datasets have real-value probabilities as targets, permitting to perform regression in log-odds space.

**Setup and Evaluation**  We defined each task to consist of a single observation. Models made predictions on the next observation immediately after finishing learning current task. Models were then evaluated with one-step-ahead absolute error[3], which is then used to compute the mean cumulative absolute error (MCAE) at time $t$: $\frac{1}{t}\sum_{i=1}^{t}|y_i^* - y_i|$ where $y^*$ is the predicted value and $y_i$ is the ground truth. We further approximated the Gaussian posterior distribution by a point mass centered around the mode. It should be noticed that for linear regression, the Laplace Propagation

---

[3]in probability space for classification tasks; in data space for regression tasks. With the exception of NBAPlayer dataset, we evaluated models with predictive log probability $\frac{1}{t}\sum_{i=2}^{t}\log p(y_i|y_{1:i-1}, x_{1:i})$.

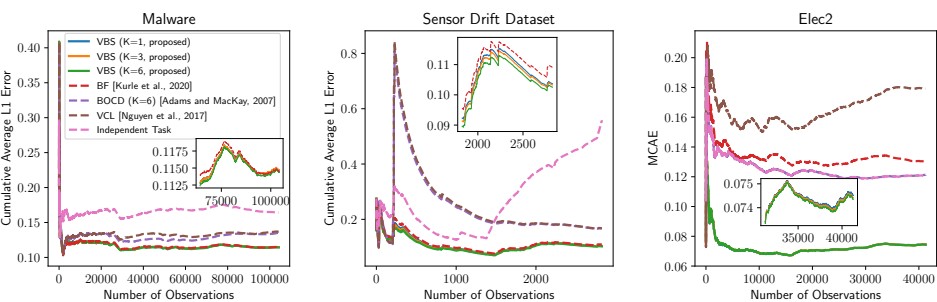

Figure 4: One-step ahead performance of the online malware detection experiments. Proposed methods outperform the baseline.

has the same mode as Variational Continual Learning, and the independent task has the same mode as its Bayesian counterpart.

**Results**   We reported the result of the dominant hypothesis of VBS with large beam size. Fig. G.2 shows MCAE over time for the first three datasets. Our methods remain lower prediction error of all time while baselines are subject to strong fluctuations or inability to adapt to distribution shifts. Another observation is that VBS with different beam sizes performed similarly in this case.

### G.2.1   Baseline Hyperparameters

**BOCD**   We only keep the top three or six most probable run length after each time step.

We tuned the hyperparameter $\lambda$ in the hazard function, or the transition probability. $\lambda^{-1}$ is searched in $\{0.1, 0.2, 0.3, 0.4, 0.5, 0.6, 0.7, 0.8, 0.9, 0.99\}$.

Malware selects $\lambda^{-1} = 0.3$; Elec2 selects $\lambda^{-1} = 0.9$; SensorDrift selects $\lambda^{-1} = 0.6$; NBAPlayer selects $\lambda^{-1} = 0.99$.

**BF**   We implemented Bayesian Forgetting according to [Kurle et al., 2020].

We tuned the hyperparameter $\beta$ as the forgetting rate such that $p(z_t|D_{t-1}) \propto p_0(z_t)^{1-\beta}q_{t-1}(z_t|D_{t-1})^{\beta}$ where $0 < \beta < 1$. $\beta$ is searched in $\{0.001, 0.01, 0.1, 0.2, 0.3, 0.4, 0.5, 0.6, 0.7, 0.8, 0.9, 0.95, 0.97, 0.98, 0.99, 0.995, 0.999\}$.

Malware selects $\beta = 0.999$; Elec2 selects $\beta = 0.98$; SensorDrift selects $\beta = 0.9$; NBAPlayer selects $\beta = 0.9$.

### G.2.2   Malware[4]

**Dataset**   There are 107856 programs collected from 2010.11 to 2014.7 in the monthly order. Each program has 482 counting features and a real-valued probability $p \in [0, 1]$ of being malware. This ground truth probability is the proportion of 52 antivirus solutions that label malware. We used the first-month data (2010.11) as the validation dataset and the remaining data as the test dataset. To enable analytic online update, we cast the binary classification problem in the log-odds space and performed Bayesian linear regression. We filled log-odds boundary values to be $-5$ and $4$, corresponding to probability 0 and 1, respectively. Our methods achieved comparable results reported in [Huynh et al., 2017] on this dataset.

**Hyperparameters**   We searched the hyperparameters $\sigma_n^2, \xi_0 = \log \frac{p_0}{1-p_0}$, and $\beta$ using the validation set. Specifically, we extensively searched $\sigma_n^2 \in \{0.1, 0.2, \cdots, 0.9, 1, 2, \cdots, 10, 20, \cdots, 100\}$, $p_0 \in \{0.5\}$, $\beta^{-1} \in \{1.01, 1.05, 1.1, 1.2, 1.5, 2, 5\}$. On most values the optimization landscape is

---

[4]`https://archive.ics.uci.edu/ml/datasets/Dynamic+Features+of+VirusShare+Executables`

monotonic and thus the search quickly converges around a local optimizer. Within the local optimizer, we performed the grid search, which focused on $\beta \in [1.05, 1.2]$.

We found all methods favored $\sigma_n^2 = 40$. And for VGS and VBS, the uninformative prior of the change variable $p_0 = 0.5$ was already a suitable one. VGS selected $\beta^{-1} = 1.2$, VBS (K=3) selected $\beta^{-1} = 1.07$, and VBS (K=6) selected $\beta^{-1} = 1.05$. Although searched $\beta^{-1}$ varies for different beam size, the performance of different beam size in this case, based on our experience, is insensitive to the varying $\beta$.

### G.2.3 SensorDrift[5]

**Dataset**  We focused on one kind of gas, *Acetaldehyde*, retrieved from the original gas sensor drift dataset [Vergara et al., 2012], which spans 36 months. We formulated an regression problem of predicting the gas concentration level given all the other features. The dataset contains 128 features and 2926 samples in total. We used the first batch data (in the original dataset) as the validation set and the others as the test set. Since the scales of the features vary greatly, we scaled each feature with the sample statistics computed from the validation set, leading to zero mean and unit standard deviation for each feature.

**Hyperparameters**  We found that using a Bayesian forgetting module (instead of tempered posterior module mentioned in the main paper), which corresponds to $s_t = 1$, works better for this dataset. Since we scaled the dataset, we therefore set the hyperparameter $\sigma_n^2 = 1$ for all methods. We searched $\xi_0 = \log \frac{p_0}{1-p_0}$ and $\beta$ using the validation set. Specifically, we did the grid search for the prior probability of change point $p \in \{0.501, 0.503, 0.505, 0.507, 0.509, 0.511, 521, 0.55, 0.6, 0.7, 0.8, 0.9\}$ and the temperature $\beta \in \{0.5, 0.6, 0.7, 0.8, 0.9\}$. The search procedure selects $p = 0.507$ and $\beta = 0.7$ for all beam size $K$.

### G.2.4 Elec2[6]

**Dataset**  The dataset contains the electricity price over three years of two Australian states, New South Wales and Victoria [Harries and Wales, 1999]. While the original problem was 0-1 binary classification, we re-produced the targets with real-value probabilities since all necessary information forming the original target is contained in the dataset. Specifically, we re-defined the target to be the probability of the price in New South Wales increasing relative to the price of the last 24 hours. Then we performed linear regression in the log-odds space. We filled log-odds boundary values to be $-4$ and $4$, corresponding to probability 0 and 1, respectively. After removing the first 48 samples (for which we cannot re-produce the targets), we had 45263 samples, and each sample comprised 14 features. The first 4000 samples were used for validation while the others were used for test.

**Hyperparameters**  We searched the hyperparameters $\sigma_n^2, \xi_0 = \log \frac{p_0}{1-p_0}$, and $\beta$ using the validation set. Specifically, we extensively searched $\sigma_n^2 \in \{0.01, 0.02, \cdots, 0.1, 0.2, \cdots, 1, 2, \cdots, 10, 20, \cdots, 100\}$, $p_0 \in \{0.5\}$, $\beta^{-1} \in \{1.05, 1.1, 1.2, 1.5, 2, 5\}$.

VCL favored $\sigma_n^2 = 0.01$, and we set this value for all other methods. VGS selected $\beta^{-1} = 1.2$. VBS (K=3) and VBS (K=6) inherited the same $\beta$ value from VGS.

### G.2.5 NBAPlayer[7]

**Dataset**  The original dataset contains part of the game logs of 2015-2016 NBA season in json files. The log records each on-court player's position (in a 2D space) at a rate of 25 Hz. We pre-processed the logs and randomly extracted ten movement trajectories for training set and another ten trajectories for test set. For an instance of the trajectory, we selected Wesley Matthews's trajectory at the 292th event in the game of Los Angeles Clippers vs. Dallas Mavericks on Nov 11, 2015. The trajectories vary in length and correspond to players' strategic movement. After extracting the trajectories, we fix

---

[5]http://archive.ics.uci.edu/ml/datasets/Gas+Sensor+Array+Drift+Dataset+at+Different+Concentrations

[6]https://www.openml.org/d/151

[7]https://github.com/linouk23/NBA-Player-Movements

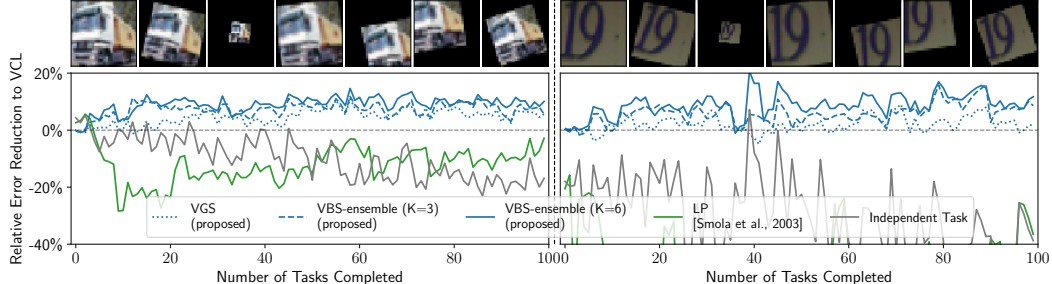

Figure 5: (**Bottom**) Running test performance of our proposed VBS and VGS algorithms compared to various baselines on transformed CIFAR-10 (left) and SVHN (right). (**Top**) Examples of transformations that we used for introducing covariate shifts.

Table 1: Convolution Neural Network Architecture

| LAYER | FILTER SIZE | FILTERS | STRIDE | ACTIVATION | DROPOUT |
|---|---|---|---|---|---|
| CONVOLUTIONAL | $3 \times 3$ | 32 | 1 | RELU | |
| CONVOLUTIONAL | $3 \times 3$ | 32 | 1 | RELU | |
| MAXPOOLING | $2 \times 2$ | | 2 | | 0.2 |
| CONVOLUTIONAL | $3 \times 3$ | 64 | 1 | RELU | |
| CONVOLUTIONAL | $3 \times 3$ | 64 | 1 | RELU | |
| MAXPOOLING | $2 \times 2$ | | 2 | | 0.2 |
| FULLYCONNECTED | | 10 | | SOFTMAX | |

the data set and then evaluate all methods with it. Specifically, we regress the current position on the immediately previous position–modeling the player's velocity.

**Hyperparameters** We searched the hyperparameters $\sigma_n^2, \xi_0 = \log \frac{p_0}{1-p_0}$, and $\beta$ using the training set. Specifically, we searched $\sigma_n^2 \in \{0.001, 0.01, 0.1, 0.5, 1.0, 10., 100.\}$ and $p \in \{0.501, 0.503, 0.505, 0.507, 0.509, 0.511, 521, 0.55, 0.6, 0.7, 0.8, 0.9\}$ and $\beta \in \{0.5, 0.6, 0.7, 0.8, 0.9\}$.

VCL favored $\sigma_n^2 = 0.1$, and we set this value for all other methods. VBS (K=1, VGS) selected $\beta = 0.5$ and $p = 0.513$. VBS (K=3) selected $p = 0.507$ and $\beta = 0.6$. VBS (K=6) selected $p = 0.505$ and $\beta = 0.7$. In generating the plots on the example trajectory, we used $p = 0.507$ with varying $\beta$ and varying beam size $K$.

### G.3 Bayesian Deep Learning Experiments

We performed the Bayesian Deep Learning experiments on a server with Intel(R) Xeon(R) Gold 5218 CPU @ 2.30GHz and Nvidia TITAN RTX GPUs. Regarding the running time, VCL, and Independent task (Bayes) takes five hours to finish training; Independent task takes three hours; VGS takes two GPUs and five hours; VBS (K=3) takes six GPUs and five hours; VBS (K=6) takes six GPUs and 10 hours. When utilizing multiple GPUs, we implemented task multiprocessing with process-based parallelism.

**Datasets with Covariate shifts** We used two standard datasets for image classification: CIFAR-10 [Krizhevsky et al., 2009] and SVHN [Netzer et al., 2011]. We adopted the original training set and used the first 5000 images in the original test set as the validation set and the others as the test set. We further split the training set into batches (or tasks in the continual learning literature) for online learning, each batch consisting of a third of the full data. Each transformation (either rotation, translation, or scaling) is generated from a fixed, predefined distribution (see below for **Transformations**) as covariate shifts. Changes are introduced every three tasks, where the total number of tasks was 100.

Table 2: Hyerparameters of Bayesian Deep Learning Models for CIFAR-10

| MODEL | LEARNING RATE | BATCH SIZE | NUMBER OF EPOCHS | $\beta$ | $\xi_0$ OR $\lambda^{-1}$ | $T$ |
|---|---|---|---|---|---|---|
| LP | 0.001 | 64 | 150 | N/A | N/A | N/A |
| BOCD | 0.0005 | 64 | 150 | N/A | 0.3 | 20000 |
| BF | 0.0005 | 64 | 150 | 0.9 | N/A | 20000 |
| VCL | 0.0005 | 64 | 150 | N/A | N/A | N/A |
| VBS | 0.0005 | 64 | 150 | 2/3 | 0 | 20000 |

Table 3: Hyperparameters of Bayesian Deep Learning Models for SVHN.

| MODEL | LEARNING RATE | BATCH SIZE | NUMBER OF EPOCHS | $\beta$ | $\xi_0$ OR $\lambda^{-1}$ | $T$ |
|---|---|---|---|---|---|---|
| LP | 0.001 | 64 | 150 | N/A | N/A | N/A |
| BOCD | 0.00025 | 64 | 150 | N/A | 0.3 | 20000 |
| BF | 0.00025 | 64 | 150 | 0.9 | N/A | 20000 |
| VCL | 0.00025 | 64 | 150 | N/A | N/A | N/A |
| VBS | 0.00025 | 64 | 150 | 2/3 | 0 | 20000 |

**Transformations**    We used Albumentations [Buslaev et al., 2020] to implement the transformations as covariate shifts. As stated in the main paper, the transformation involved rotation, scaling, and translation. Each transformation factor followed a fixed distribution: rotation degree conformed to $\mathcal{N}(0, 10^2)$; scaling limit conformed to $\mathcal{N}(0, 0.3^2)$; and the magnitude of vertical and horizontal translation limit conformed to Beta$(1, 10)$, and the sampled magnitude is then rendered positive or negative with equal probability. The final scaling and translation factor should be the corresponding sampled limit plus 1, respectively.

**Architectures and Protocol**    All Bayesian and non-Bayesian methods use the same neural network architecture. We used a truncated version of the VGG convolutional neural network (in Table 1) on both datasets. We confirmed that our architecture achieved similar performance on CIFAR10 compared to the results reported by Zenke et al. [2017] and Lopez-Paz and Ranzato [2017] in a similar setting. We implemented the Bayesian models using TensorFlow Probability and the non-Bayesian counterpart (namely Laplace Propagation) using TensorFlow Keras. Every bias term in all the models were treated deterministically and were not affected by any regularization.

We initialize each algorithm by training the model on the full, untransformed dataset. During every new task, all algorithms are trained until convergence.

**Tempered Conditional ELBO**    In the presence of massive observations and a large neural network, posterior distributions of change variables usually have very low entropy because of the very large magnitude of the difference between conditional ELBOs as in Eq. 6. Therefore change variables become over confident about the switch-state decisions. The situation gets even more severe in beam search settings where almost all probability mass is centered around the most likely hypothesis while the other hypotheses get little probability and thereby will not take effect in predictions. A possible solution is to temper the conditional ELBO (or the marginal likelihood) and introduce more uncertainty into the change variables. To this end, we divide the conditional ELBO by the number of observations. It is equivalent to set $T = 20000$ in Eq. 6. This practice renders every hypothesis effective in beam search setting.

**Hyperparameters, Initialization, and Model Training**    The hyperparameters used across all of the models for the different datasets are listed in Tables 2 and 3. Regarding the model-specific parameters, we set $\xi_0$ to 0 for both datasets and searched $\beta$ in the values $\{5/6, 2/3, 1/2, 1/4\}$ on a validation set. We used the first 5000 images in the original test set as the validation set, and the others as the test set. We found that $\beta = 2/3$ performs relatively well for both data sets. Optimization parameters, including learning rate, batch size, and number of epochs, were selected to have the best validation performance of the classifier on one independent task. To estimate the change variable $s_t$'s

variational parameter, we approximated the conditional ELBOs 7 by averaging 10000 Monte Carlo samples.

As outlined in the main paper, we initialized each algorithm by training the model on the full, untransformed dataset. The model weights used a standard Gaussian distribution as the prior for this meta-initialization step.

When optimizing with variational inference, we initialized $q(\mathbf{z}_t)$ to be a point mass around zero for stability. When performing non-Bayesian optimization, we initialized the weights using Glorot Uniform initializer [Glorot and Bengio, 2010]. All bias terms were initialized to be zero.

We performed both the Bayesian and non-Bayesian optimization using ADAM [Kingma and Ba, 2015]. For additional parameters of the ADAM optimizer, we set $\beta_1 = 0.9$ and $\beta_2 = 0.999$ for both data sets. For the deep Bayesian models specifically, which include VCL and VBS, we used stochastic black box variational inference [Ranganath et al., 2014, Kingma and Welling, 2014, Zhang et al., 2018]. We also used the Flipout estimator [Wen et al., 2018] to reduce variance in the gradient estimator.

**Predictive Distributions** We evaluated the most likely hypothesis' predictive posterior distribution of the test set by the following approximation:

$$p(\mathbf{y}^*|\mathbf{x}^*, \mathcal{D}_{1:t}, s_{1:t}) \approx \frac{1}{N} \sum_{n=1}^{N} p(\mathbf{y}^*|\mathbf{x}^*, \mathbf{z}_{s_{1:t}}^{(n)})$$

where $N$ is the number of Monte Carlo samples from the variational posterior distribution $q^*(\mathbf{z}_t|s_{1:t})$. In our experiments we found $S = 10$ to be sufficient. We take $\arg\max_{\mathbf{y}_t} p(\mathbf{y}^*|\mathbf{x}^*, , \mathcal{D}_{1:t})$ to be the predicted class.

LP only used the MAP estimation $\mathbf{z}_t^*$ to predict the test set: $p(\mathbf{y}^*|\mathbf{x}^*, \mathcal{D}_{1:t}) \approx p(\mathbf{y}^*|\mathbf{x}^*, , \mathbf{z}_t^*)$.

**Standard Deviation in the Main Text Table 1** The results in this table were summarized and reported by taking the average over tasks. Each algorithm's confidence, which is usually evaluated by computing the standard deviation across tasks in stationary environments, now is hard to evaluate due to the non-stationary setup. These temporal image transformations will largely affect the performance, leaving the blindly computed standard deviation meaningless since the standard deviation across all tasks represents both the data transformation variation and the modeling variation. To evaluate the algorithm's confidence, we proposed a three-stage computation. We first segment the obtained performance based on the image transformations (in our case, we separate the performance sequence every three tasks). Then we compute the standard deviation for every performance segment. Finally, we average these standard deviations across segments as the final one to be reported. In this way, we can better account for the data variation in order to isolate the modeling variation.

**Running Performance.** We also reported the running performance for both our methods and some baselines for each task over time (100 tasks in total) in Fig. 5. In the bottom panel, to account for varying task difficulties, we show the percentage of the error reduction relative to VCL, a Bayesian online learning baseline. Our proposed approach can achieve 10% error reduction most of the time on both datasets, showing the adaptation advantage of our approach. The effect of beam search is also evident, with larger beam sizes consistently performing better. The top panel shows some examples of the transformations that we used for introducing covariate shifts manually.

### G.4 Dynamic Word Embeddings Experiments

We performed the Dynamic Word Embeddings experiments on a server with Intel(R) Xeon(R) Gold 5218 CPU @ 2.30GHz and Nvidia TITAN RTX GPUs. Regarding the running time, for qualitative experiments, Google Books and Congressional Records take eight GPUs and about 24 hours to finish; UN Debates take eight GPUs and about 13 hours to finish. For quantitative experiments, since the vocabulary size and latent dimensions are smaller, each model corresponding to a specific $\xi_0$ takes eight GPUs and about one hour to finish. When utilizing multiple GPUs, we implemented task multiprocessing with process-based parallelism.

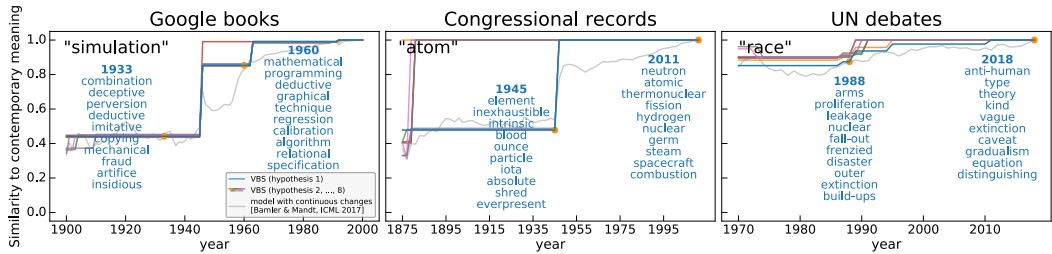

Figure 6: Dynamic Word Embeddings on Google books, Congressional records, and UN debates, trained with VBS (proposed, colorful) vs. VCL (grey). In contrast to VCL, VBS reveals sparse, time-localized semantic changes (see main text).

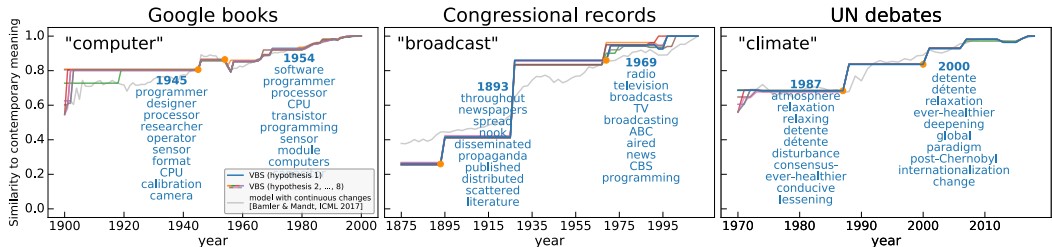

Figure 7: Additional results of Dynamic Word Embeddings on Google books, Congressional records, and UN debates.

**Data and Preprocessing**   We analyzed three large time-stamped text corpora, all of which are available online. Our first dataset is the Google Books corpus [Michel et al., 2011] consisting of $n$-grams, which is sufficient for learning word embeddings. We focused on the period from 1900 to 2000. To have an approximately even amount of data per year, we sub-sampled 250M to 300M tokens per year. Second, we used the Congressional Records data set [Gentzkow et al., 2018], which has 13M to 52M tokens per two-year period from 1875 to 2011. Third, we used the UN General Debates corpus [Jankin Mikhaylov et al., 2017], which has about 250k to 450k tokens per year from 1970 to 2018. For all three corpora, the vocabulary size used was 30000 for qualitative results and 10000 for quantitative results. We further randomly split the corpus of every time step into training set (90%) and heldout test set (10%). All datasets, Congressional Records[8], UN General Debates[9], and Google Books[10] can be downloaded online.

We tokenized Congressional Records and UN General Debates with pre-trained Punkt tokenizer in NLTK[11]. We constructed the co-occurence matrices with a moving window of size 10 centered around each word. Google books are already in Ngram format.

**Model Assumptions**   As outlined in the main paper, we analyzed the semantic changes of individual words over time. We augmented the probabilistic models proposed by Bamler and Mandt [2017] with our change point driven informative prior (Eq. 3 in the main paper) to encourage temporal sparsity.

---

[8] https://data.stanford.edu/congress_text

[9] https://dataverse.harvard.edu/dataset.xhtml?persistentId=doi:10.7910/DVN/0TJX8Y

[10] http://storage.googleapis.com/books/ngrams/books/datasetsv2.html

[11] https://www.nltk.org/

Table 4: Hyperparameters of Dynamic Word Embedding Models

| CORPUS | VOCAB | DIMS | $\beta$ | LEARNING RATE | EPOCHS | $\xi_0$ | BEAM SIZE (K) | $T$ |
|---|---|---|---|---|---|---|---|---|
| GOOGLE BOOKS | 30000 | 100 | 0.5 | 0.01 | 5000 | -10 | 8 | 1 |
| CONGRESSIONAL RECORDS | 30000 | 100 | 0.5 | 0.01 | 5000 | -10 | 8 | 1 |
| UN DEBATES | 30000 | 20 | 0.25 | 0.01 | 5000 | -1 | 8 | 1 |

Table 5: $\xi_0$ of Document Dating Tasks

| CORPUS | $\xi_0$ |
|---|---|
| GOOGLE BOOKS | -1000000, -100000, -5120, -1280, -40 |
| CONGRESSIONAL RECORDS | -100000, -1280, -320, -40 |
| UN DEBATES | -128, -64, -32, -4 |

We pre-trained the *context* word embeddings[12] using the whole corpus, and kept them constant when updating the *target* word embeddings. This practice denied possible interference on one target word embedding from the updates of the others. If we did not employ this practice, the spike and slab prior on word $i$ would lead to two branches of the "remaining vocabulary" (embeddings of the remaining words in the vocabulary), conditioned either on the spike prior of word $i$ or on the slab prior. This hypothetical situation gets severe when every word in the vocabulary can take two different priors, thus leading to exponential branching of the sequences of inferred change points. When this interference is allowed, the exponential scaling of hypotheses translates into exponential scaling of possible word embeddings for a single target word, which is not feasible to compute for any meaningful vocabulary sizes and number of time steps. To this end, while using a fixed, pre-trained context word embeddings induces a slight drop of predictive performance, the computational efficiency improves tremendously and the model can actually be learned.

**Hyperparameters and Optimization** Qualitative results in Figure 6 in the main paper were generated using the hyperparameters in Table 4. The initial prior distribution used for all latent embedding dimensions was a standard Gaussian distribution. We also initialized all variational distributions with standard Gaussian distributions. For model-specific hyperparameters $\beta$ and $\xi_0$, we first searched the broadening constant $\beta$ to have the desired jump magnitude observed from the semantic trajectories mainly for medium-frequency words. We then tuned the bias term $\xi_0$ to have the desired change frequencies in general. We did the searching for the first several time steps. We performed the optimization using black box variational inference and ADAM. For additional parameters of ADAM optimizer, we set $\beta_1 = 0.9$ and $\beta_2 = 0.999$ for all three corpora. In this case, we did **not** temper the conditional ELBO by the number of observations (correspondingly, we set $T = 1$ in Eq. 6 in the main paper).

Quantitative results of VGS in Figure 2 (c) in the main paper were generated by setting a smaller vocabulary size and embedding dimension, 10,000 and 20, respectively for all three corpora. We used an eight-hypothesis (K=8) VBS to perform the experiments. Other hyperparameters were inherited from the qualitative experiments except $\xi_0$, whose values used to form the rate-distortion curve can be found in Table 5. We enhanced beam diversification by dropping the bottom two hypotheses instead of the bottom third hypotheses before ranking. On the other hand, the baseline, "binning", and had closed-form performance if we assume (i) a uniformly distributed year in which a document query is generated, (ii) "binning" perfectly locates the ground truth episode, and (iii) the dating result is uniformed distributed within the ground truth episode. The $L1$ error associated with "binning" with episode length $L$ is $\mathbb{E}_{t \sim \mathcal{U}(1,L), t' \sim \mathcal{U}(1,L)}[|t - t'|] = \frac{L-1}{2}$. By varying $L$, we get binning's rate-distortion curve in Figure 2 (c) in the main paper.

**Predictive Distributions** In the demonstration of the quantitative results, i.e., the document dating experiments, we predicted the year in which each held-out document's word-word co-occurrence statistics $\mathbf{x}$ have the highest likelihood and measured $L1$ error. To be specific, for a given document in year $t$, we approximated its likelihood under year $t'$ by evaluating $\frac{1}{|V|} \log p(\mathbf{x}_t|\mathbf{z}_{t'}^*)$, where $\mathbf{z}_{t'}^*$ is the mode embedding in year $t'$ and $|V|$ is the vocabulary size. We predicted the year $t^* = \arg\max_{t'} \frac{1}{|V|} \log p(\mathbf{x}_t|\mathbf{z}_{t'}^*)$. We then measured the $L1$ error by $\frac{1}{T} \sum_i^T |t_i - t_i^*|$ given $T$ truth-prediction pairs.

---

[12]We refer readers to [Mikolov et al., 2013, Bamler and Mandt, 2017] for the difference between target and context word embeddings.

### G.4.1 Additional Results

**Qualitative Results**  As outlined in the main paper, our qualitative result shows that the information priors encoded with change point detection is more interpretable and results in more meaningful word semantics than the diffusion prior of [Bamler and Mandt, 2017]. Here we provide a more detailed description of the results with more examples. Figure 6 shows three selected words ("simulation", "atom", and "race"–one taken from each corpus) and their nearest neighbors in latent space. As time progresses the nearest neighboring words change, reflecting a semantic change of the words. While the horizontal axis shows the year, the vertical axis shows the cosine distance of the word's embedding vector at the given year to its embedding vector in the last available year.

The plot reveals several interpretable semantic changes of each word captured by VBS. For example, as shown by the most likely hypothesis in blue for the Congressional Records data, the term "atom" changes its meaning from "element" to "nuclear" in 1945–the year when two nuclear bombs were detonated. The word "race" changes from the cold-war era "arms"(-race) to its more prevalent meaning after 1991 when the cold war ended. The word "simulation" changes its dominant context from "deception" to "programming" with the advent of computers. The plot also showcases various possible semantic changes of all eight hypotheses, where each hypothesis states various aspects.

Additional qualitative results can be found in Figure 7. it, again, reveals interpretable semantic changes of each word: the first change of "computer" happens in 1940s–when modern computers appeared; "broadcast" adopts its major change shortly after the first commercial radio stations were established in 1920; "climate" changes its meaning at the time when Intergovernmental Panel on Climate Change (IPCC) was set up, and when it released the assessment reports to address the implications and potential risks of climate changes.

**Quantitative Results and Baseline**  Figure 2 (c) in the main paper shows the results on the three corpora data, where we plot the document dating error as a function of allowed changes per year. For fewer allowed semantic changes per year, the dating error goes up. Lower curves are better.

Now we describe how the baseline "Binning" was constructed. We assumed that we had separate word embeddings associated with episodes of $L$ consecutive years. For $T$ years in total, the associated memory requirements would be proportional to $V * T/L$, where $V$ is the vocabulary size. Assuming we could perfectly date the document up to $L$ years results in an average dating error of $\frac{L}{2}$. We then adjusted $L$ to different values and obtained successive points along the "Binning" curve.