# OpenReview forum: "Detecting and Adapting to Irregular Distribution Shifts in Bayesian Online Learning"
_NeurIPS.cc/2021/Conference — NeurIPS 2021 Poster_

### Official Review · Reviewer_kxgB · 2021-06-29

**Rating:** 8
**Confidence:** 4

**Summary:**

This submission relates a method for adapting to distribution shifts online in a Bayesian fashion. A distribution shift is represented as a binary latent variable indicating that a shift has, or has not occurred. If a shift is inferred to have occurred, then the variance of the posterior is increased before the next step of the Bayesian recursion, weakening the influence of previous observations. Computational constraints associated with maintaining many possible hypotheses of the incidence of distribution shifts are remedied by the use of beam search.

This paper is timely and well-written. The topic of online adaptation to distribution shift is very relevant to the NeurIPS community, and the submission has developed through several rounds of peer review into a mature contribution to the literature. There are some substantial limitations that would be important to address in future work, but I am happy to recommend acceptance and willing to defend my evaluation.

UPDATE 08/17/2021
The author response has addressed all my concerns. In particular it is very good to know that the full changepoint history need not be maintained, and that the method will automatically adapt, in a sense, to a poor choice of \beta. I think this is a very good paper, and have increased my score to 8.

**Ethical Concerns:**

I have no ethical concerns with this work.

**Limitations And Societal Impact:**

There are some conceptual shortcomings of the method as proposed that prevent this good paper from being great.

While the method is an online algorithm, it is not a streaming algorithm, and in practice streaming algorithms are necessary for deployment to most devices (for example, the sampling rate of a smartphone accelerometer is ~200 Hz [4]). The method is unlikely to be used in the (presumably low-latency) contexts of “... systems such as self-driving cars, robots, and financial trading algorithms…” until it is a streaming algorithm. In future work, it would be very intriguing to see if the method could be adapted to constant-time streaming sparse GPs, such as [1] or [2].

Along similar lines, the need to tune the hyperparameters offline is also a pretty serious limitation. In particular I see no reason why the \beta that worked well for one type of distribution shift would work well for another. The method could more plausibly be deployed if there was a sensible way to adapt the hyperparameters online.

The final limitation relates to the choice to model distribution shifts as binary changepoints. In particular, it would have been very interesting to see some experiments or discussion of the virtues of the changepoint detection class of methods vs. directly modeling possible shifts in the data generating process. As an example, in the rotated MNIST experiment, instead of modeling discrete (unknown) shifts, one could instead take the latent variable to be the average rotation (a continuous value). Obviously it would complicate the beam search process, but it would not be impossible, because one can always discretize a continuous signal (see e.g., the quantization scheme in [3]). I suspect that a well-specified hypothesis class of possible shifts would improve the speed of adaptation and allow for continuous shifts, which are an important case to consider (e.g. the relatively continuous shift in light intensity from day to night as a function of time).

Since this work relates to a very general class of methods, I have no concerns about potential social impact.


**Main Review:**

The approach as described is very thoroughly developed. As noted in the submission meta-data, this is the third round of peer review that this work has received, and I believe it represents a worthy contribution in its current state. There are a couple relevant references that have been omitted, but overall the literature review is adequate. I find the experiment evaluation sufficient to verify the quality of the idea.

The asymptotic memory and compute complexity (particularly in terms of the time horizon) do not appear to be clearly stated, aside from a brief discussion of memory requirements in G.4.1.
I understand that beam search remedies the combinatorial explosion of possible histories, but it does not address the fact that even a fixed set of K sequences must eventually grow to infinite length in the current formalism. Of course the histories could simply be truncated, but that approach introduces history length as another hyperparameter to tune. Is there a more principled way to address the scaling with the time horizon (see limitations for more discussion on the importance of this question)?

The way the method is introduced in terms of sufficient statistics creates the expectation that the method will be generally developed for any exponential conjugate prior. This extension would be worth considering for future work.

---- References ----

[1] Bui, T. D., Nguyen, C. V., and Turner, R. E. (2017). Streaming sparse Gaussian process approximations. In Advances in Neural Information Processing Systems 31, pages 3301–3309, Long Beach, California, USA. Curran Associates Inc.

[2] Stanton, S., Maddox, W., Delbridge, I., & Wilson, A. G. (2021, March). Kernel Interpolation for Scalable Online Gaussian Processes. In International Conference on Artificial Intelligence and Statistics (pp. 3133-3141). PMLR.

[3] Oord, A. V. D., Dieleman, S., Zen, H., Simonyan, K., Vinyals, O., Graves, A., ... & Kavukcuoglu, K. (2016). Wavenet: A generative model for raw audio. arXiv preprint arXiv:1609.03499.

[4] Solin, A., Hensman, J., & Turner, R. E. (2018). Infinite-horizon Gaussian processes. arXiv preprint arXiv:1811.06588.


**Time Spent Reviewing:**

2.5

---

> ### Author Response · Authors · 2021-08-10
> **Response to Reviewer kxgB**
>
> We thank the reviewer for their insightful review and positive reception of our work. We address the comments and questions below.
>
> > “There are a couple of relevant references that have been omitted...”
>
>   Thanks for pointing out these references. We will make sure to discuss and cite them.
>
> > “The asymptotic memory and compute complexity (particularly in terms of the time horizon) do not appear to be clearly stated” “...but it does not address the fact that even a fixed set of K sequences must eventually grow to infinite length in the current formalism...”
>
> Thanks for bringing up these important points. We have added a comment on the complexity in the general comment area in our response. Our short answer is that our space and time complexity is linear in K and independent of the sequence length T, despite the fact that we currently label each hypothesis by its associated changepoint sequence. We only need to save the changepoint history if we are explicitly interested in it; i.e., we can run the algorithm as a streaming algorithm in the sense that its time and space complexity is independent of T. In the final paper we will discuss this more clearly in the corresponding technical section and add a discussion.
>
> > “The way the method is introduced in terms of sufficient statistics creates the expectation that the method will be generally developed for any exponential conjugate prior. ”
>
> Yes, the method could be generalized by reading off sufficient statistics of the previous approximate posterior. To this end, we need a sufficient statistic that is associated with some measure of entropy or variance that we broaden after each detected change. For example, the Gamma distribution can broaden its scale, and for the categorical distribution, we can increase its entropy/temperature. We will add a corresponding comment to the paper and consider adopting the more general exponential family notation.
>
> > “While the method is an online algorithm, it is not a streaming algorithm...”
>
> Thanks for your comment: we believe our approach would qualify as a streaming algorithm, with the understanding that a streaming algorithm would have space and time complexity that does not depend on the sequence length T. See also our earlier response to the complexity and the “general comments” above. We only need to store the changepoint history if we are explicitly interested in it (as opposed to just adapting the underlying model to distribution shifts).
>
> > “Along similar lines, the need to tune the hyperparameters offline is also a pretty serious limitation. In particular, I see no reason why the \beta that worked well for one type of distribution shift would work well for another. The method could more plausibly be deployed if there was a sensible way to adapt the hyperparameters online.”
>
> Thanks for raising this interesting point. Although we agree that it would be desirable to get rid of the hyperparameter beta, our algorithm has a certain level of robustness to compensate for poor choices of beta. If beta is chosen to be too small (i.e., a single tempering step would erase too much information from the posterior), our algorithm will in general postulate fewer changepoints to compensate; conversely, if beta is too close to one (i.e., too much memory is kept), our algorithm will compensate by using multiple tempering steps in sequence. This behavior can be qualitatively seen in Fig. 2 (middle) on the basketball trajectory dataset. We will add a corresponding discussion.
>
> We agree that learning beta on the fly is an interesting idea for future research (e.g., one could place an exponential distribution as a prior over beta and infer it); however, this is currently outside of the scope of the paper.
>
> > “The final limitation relates to the choice to model distribution shifts as binary changepoints. In particular, it would have been very interesting to see some experiments or discussion of the virtues of the changepoint detection class of methods vs. directly modeling possible shifts in the data generating process.”
>
> This is a good point; our approach could certainly do this. Our proposed version is a more agnostic approach for generic shifts. If we had more domain-dependent information, we could incorporate modeling data transformations into our method. We will discuss this aspect in the discussion section of our paper.

---

> > ### Comment · Reviewer_kxgB · 2021-08-18
> > **Thank you for the response**
> >
> > Thanks for your detailed response! I do not have any additional concerns.

---

### Official Review · Reviewer_qHGH · 2021-07-16

**Rating:** 7
**Confidence:** 3

**Summary:**

This paper addresses the topic of changepoint detection in the event of distribution shift. A Bayesian model meant to detect such changes online, and able to discount historical information after a distributional shift, is formulated to express the problem. Algorithmic implementation is provided and evaluated across a number of data sets.

**Limitations And Societal Impact:**

Short and sufficient considerations on broader impact are offered at the end of the paper.

**Main Review:**

The paper is well organized and written: the problem is properly defined, assumptions are clearly stated, formalization rigorous; the reasoning steps in the development of the inference algorithm are also easy to follow. Experiments are extensive, and the sub-selection in the main paper suitably illustrative; some considerations on the compuational cost (as a function of the variational hyperparameters like beam size) against the SotA methods considered would have been interesting. Related works clearly relates the current work to the literature, although this reviewer is not able to judge independently how innovative the work is within the context of changepoint detection models.

**Time Spent Reviewing:**

3

---

> ### Author Response · Authors · 2021-08-10
> **Response to Reviewer qHGH**
>
> We thank the reviewer for the helpful review.
>
> > “...some considerations on the computational cost (as a function of the variational hyperparameters like beam size) against the SotA methods considered would have been interesting.”
>
> Thanks for your suggestion; this is a valid point issued by multiple reviewers. We have added a comment on the complexity in the general comment area of our response. Note also that our paper already has a brief discussion of the computational cost in L217 which we will expand. Our algorithm’s complexity for both time and space is *linear* in the beam size K, and constant in the sequence length T. As such, its computational cost is approximately 2K times larger than the baselines Bayesian Forgetting (BF) and Variational Continual Learning (VCL). Note that K is typically small, e.g., K=3 or 6 for most of our experiments. The number of variational parameters to optimize is twice the number of model parameters (Gaussian means and variances). We will expand the discussion on the scaling and costs of our approach in the final version.

---

### Official Review · Reviewer_GfAk · 2021-07-18

**Rating:** 4
**Confidence:** 4

**Summary:**

In this paper, the authors propose a method to jointly model the data and also change points in the model parameters.
This is accomplished following a Bayesian approach, where the full vector of change point detections (as binary indicators) is part of the model, and it informs the marginal posterior.
The method is illustrated in an interesting mix of real and synthetic data.

UPDATE: The authors' responses were succinct and useful. However, I still think that the paper relies on too many approximations/heuristics. The approach is sensible, and the paper could be impactful, but they don't constitute a Neurips-level contribution.

**Limitations And Societal Impact:**

.

**Main Review:**

In this paper, the authors propose a method to jointly model the data and also change points in the model parameters.
This is accomplished following a Bayesian approach, where the full vector of change point detections (as binary indicators) is part of the model, and it informs the marginal posterior.
The method is illustrated in an interesting mix of real and synthetic data.

Overall, this paper is well written and deals with an important problem. However, I have some important concerns:

* The method looks highly impractical to me. A full vector of change points is being carried around, and so the dimensionality of the model explodes. There are some interesting heuristics proposed, but they are all approximate heuristics, and so it is unclear how all this works in practice.
e.g., how to approximate a bimodal distribution with a unimodal one ? ( L182).

* On a related note, I find the normality assumption in (2) and (3) to be severely restrictive. I understand that the Bayesian approach requires this kind of simplifications, but this assumption automatically precludes practical cases. with, say, categorical or sparse or missing data.

* At several points the authors describe an "exact inference" scheme. I am not sure what exact inference means here, especially given the number of heuristic methods involved.

* In L296, what does it mean to reproduce the targets with real-valued probabilities?



**Time Spent Reviewing:**

3

---

> ### Author Response · Authors · 2021-08-10
> **Response to Reviewer GfAk**
>
> We thank the reviewer for the detailed review. We hope that we can convince the reviewer that some of the concerns are potential misunderstandings.
>
>
> > “The method looks highly impractical to me. A full vector of change points is being carried around, and so the dimensionality of the model explodes. ”
>
> Without approximations, it is indeed impractical to keep track of all possible changepoint hypotheses since the number of hypotheses would grow combinatorially - this is a fundamental challenge in all changepoint detection algorithms with unknown numbers of changepoints. This issue is exactly the reason why we proposed our beam search (and greedy) approximations. In our approach, the number of changepoint hypotheses kept in memory is restricted to a fixed number K. For every discrete time step t = 0,...,T, the possibilities of change vs. no change (s_t = 0/1) are both evaluated (leading to a temporary doubling of the number of hypotheses from K to 2K), and the probabilities are computed, re-ranked, and the *least likely* K of the possible 2K hypotheses are discarded. In this way, our algorithm operates with constant costs and space (and time) proportional to K (see also “General Comments” above). Also, note that the change point history doesn’t have to be stored so that the algorithm is O(1) in the sequence length T.
>
> > “... it is unclear how all this works in practice. e.g., how to approximate a bimodal distribution with a unimodal one? (L182)”
>
> Thanks for your question. The truncation procedure from a bimodal distribution to a unimodal one is accomplished by the greedy search step 3 (see L188): we approximate the bimodal distribution with the unimodal component that has a higher likelihood. Such uni-modal posterior approximations are frequently used in machine learning. For example, maximum likelihood and maximum a posteriori estimation methods amount to approximating a multi-modal posterior by a (uni-modal) point estimate. In our Bayesian approach, we go beyond these methods by using a Gaussian posterior approximation. The truncation of the posterior is a practical necessity to avoid a combinatorial explosion of posterior modes and is a widely used (and successful) strategy in posterior approximations for changepoint and mixture model problems [Adams and MacKay, 2007, Murphy, 2012, Barber, 2012].
>
> > “I find the normality assumption in (2) and (3) to be severely restrictive … (which) precludes categorical or sparse or missing data.”
>
> We would like to clarify a potential misunderstanding: unimodal *posterior* approximations do not imply unimodal modeling assumptions for the data (see also our previous response). Gaussian approximations for posterior distributions on parameters are widely used in continual learning [Kirkpatrick et al., 2017, Nguyen et al., 2018, Bamler and Mandt, 2017]. Note that the *output distributions* (modeling the data) can still be multi-modal and non-Gaussian, e.g., categorical, as we show in our dynamic word embedding experiments. Missing data is an additional level of complexity that goes beyond the scope of the present work, but is possible to address as well in deep latent variable models with Gaussian posteriors, see e.g. [Fortuin et al, 2020].
>
> > “...what exact inference means here, especially given the number of heuristic methods involved.”
>
> This is a helpful comment that we will use to make our approach more accessible. “Exact inference” refers to exact Bayesian inference, i.e., being able to exactly specify the posterior distribution on parameters (e.g., via closed-form equations). ​​In many cases, however, the functional form of the posterior cannot be computed directly (e.g., the normalization constant is intractable) and approximation methods such as variational inference are widely used to produce a tractable approximation to the exact (but intractable to compute) posterior. As you correctly mention, this approach still involves heuristic approximations (extracting sufficient statistics from the previous time step’s posterior), which we will write out more clearly.
>
>
> > “In L296, what does it mean to reproduce the targets with real-valued probabilities?”
>
> Thanks for your question; this refers to a data modeling choice we made for a particular dataset. Our Elec2 dataset contains a collection of features and multiple (possibly disagreeing) labels that were generated by different malware detectors. This set of labels can either be binarized (using a majority vote) and treated as a binary classification problem, or converted into a probability (by averaging) and treated as a regression problem. We chose the regression option where our target variable is the ensemble probability score. We will change our wording accordingly.
>
> [Adams and MacKay, 2007] Adams, Ryan Prescott, and David JC MacKay. "Bayesian online changepoint detection." arXiv preprint arXiv:0710.3742 (2007).
>
> [Fortuin et al, 2020] GP-VAE: Deep Probabilistic Time Series Imputation. V. Fortuin et al. Artificial Intelligence and Statistics (AISTATS 2020).
>
> [Kirkpatrick et al., 2017] Kirkpatrick, James, et al. "Overcoming catastrophic forgetting in neural networks." Proceedings of the National Academy of Sciences 114.13 (2017): 3521-3526.
>
> [Nguyen et al., 2018] Nguyen, Cuong V., et al. "Variational Continual Learning." International Conference on Learning Representations. 2018.
>
> [Bamler and Mandt, 2017] Bamler, Robert, and Mandt, Stephan. "Dynamic word embeddings." International Conference on Machine Learning. PMLR, 2017.
>
> [Murphy, 2012] Murphy, Kevin P. Machine learning: a probabilistic perspective, p649-662. MIT press, 2012.
>
> [Barber, 2012] Barber, David. Bayesian reasoning and machine learning, p521-536. Cambridge University Press, 2012.

---

### Official Review · Reviewer_cuE8 · 2021-07-28

**Rating:** 7
**Confidence:** 2

**Summary:**

The paper focuses on distribution shift, i.e. the distribution that generates the data changes with time, something that can cause problems to static models since now the train and test distributions are not the same. The authors take an online Bayesian perspective and try to detect a changes in the distribution based on the data points that are received in each iteration. They also perform extensive experiments in different domains.


**Main Review:**

Overall, I liked the paper, it provided an interesting perspective in distribution shift detection which has become a core problem to study recently in machine learning literature. The presentation of the paper is clear, I liked that the authors started by presenting the exact inference, before the introduction of variational inference, it helped understand the main idea although some parts of the paper were still technical and hard to follow in detail. The experiments were also very thorough and in several different datasets. One question I have for the experimental part, is how does the VBS approach compare with just using static models. In the case of CIFAR for example there is a large performance gap between the best performance that a neural network can achieve for normal CIFAR and the accuracy that VBS obtains for the modified dataset used here. I am wondering how well does a static model trained on CIFAR performs for the modified dataset and how does it compare to VBS. I am also not sure that I understand what the "trivial baseline" of independent batches is. Does it mean that you use the data points from this batch to learn a model? How does this work for CIFAR when you have only 100 samples per batch? (I probably misunderstood that but I would appreciate the clarification)

Typos:
- l320: MNIST or SVHN: should it be CIFAR?
- l347: whilee

**Time Spent Reviewing:**

5

---

> ### Author Response · Authors · 2021-08-10
> **Response to Reviewer cuE8**
>
> We thank the reviewer for the constructive feedback and address the questions below.
>
> > “In the case of CIFAR [...] there is a large performance gap between the best performance that a neural network can achieve [...] and VBS [...]. How well does a static model trained on CIFAR perform [...]?”
>
> Good question. We didn’t test a static model (trained on CIFAR) as a baseline, because it would not accurately reflect the challenges of our setup. The fundamental problem we study is that data is revealed sequentially and with distribution shifts. A fair baseline would instead be a “static” model that sees the data sequentially and in the same order as the other models, without modeling distribution shifts. In some sense, VCL is already such a baseline.
>
> However, we will adopt your suggestion of training a static model on CIFAR and testing it on our data--not as a baseline, but as a means to assess the overall difficulty level of our data set. We will add these results to the final version.
>
> > “Not sure that I understand what the ‘trivial baseline’ of independent batches is … How does this work for CIFAR when you have only 100 samples per batch?”
>
> We agree that we could have explained this more clearly and acknowledge the need to update and clarify our setup more. In particular, we differentiate between “batch” (a subset of data coming from the same distribution) and “mini-batch” (for gradient-based learning). While our algorithms use a mini-batch size of 100, our “batch” size is 20,000 in our CIFAR and SVHN experiments, hence a third of these data sets. The “independent batch” baseline is trivial because it doesn’t try to use information from past batches, but instead just retrains a model from scratch for each batch (of 20,000 data points). We will make sure in our revision to clearly communicate the distinction between “batch” and “mini-batch” at the beginning of the experiments section.

---

> > ### Comment · Reviewer_cuE8 · 2021-08-28
> > **Response to Authors**
> >
> > Thank you for the response, that clarifies it. I will keep my score.

---

### Author Response · Authors · 2021-08-10
**General comments**

We thank all reviewers for their valuable feedback. Before we respond to each reviewer individually, we comment on reviewer questions that occurred across multiple reviews. We will also address these issues in the final version of our paper.

1. **Our algorithm’s complexity.** A shared question was how our algorithm scales in time and space, in particular as a function of the sequence length T.

    Our paper already contains a brief discussion of the algorithm’s complexity as a function of the beam size K in line 217. Both time and space scale *linearly* with K. As such, its computational cost is only about 2K times larger than the baselines of Bayesian Forgetting (BF) and Variational Continual Learning (VCL). Note that K is typically small, e.g., K=3 or 6 for most of our experiments.

    Notably, our algorithm’s complexity in time and space is O(1) in the sequence length T. This may be not obvious from the notation where we use change point sequences $s_{1:T}$ to indicate each hypothesis, but note that these are just used as symbols that make our notation convenient. The only exception to this scaling would be for an application asking for the most likely changepoint sequence in hindsight. In this case, the changepoint sequence (but not the associated model parameters) would need to be stored, incurring a cost of storing exactly K x T binary variables. This is, however, not necessary when the focus is only on adapting to distribution shifts.

2. **Beyond Gaussian posterior approximations.** Two reviewers asked about the need to carry-out Gaussian posterior approximations. While different posterior approximations could be used, we chose the Gaussian approximation since it is simple and is widely used (and broadly effective) in practice in Bayesian inference [e.g., Murphy, 2012, pp.649-662]. Extensions to exponential families should also be possible, as we discuss in the individual responses. More complex (e.g. multimodal) possible alternatives for posterior approximation include, for example, Gaussian mixtures with a small number of mixture components, but with potentially diminishing returns in terms of the tradeoff between approximation accuracy and complexity. The main purpose of such approximations is to avoid an exponential growth of hypotheses over time.

[Murphy, 2012] Murphy, Kevin P. Machine Learning: a Probabilistic Perspective, MIT Press, 2012.

---

### Decision · Program_Chairs · 2021-09-27

**Decision:**

Accept (Poster)

**Comment:**

This paper proposed a Bayesian framework for detecting and adaptive to irregular distribution shifts. The reviewers generally appreciated the submission, and felt that the author response did a good job of addressing questions. While there were still some remaining questions about design decisions in the approach (see reviews), on the whole the paper represents a good contribution. Please carefully consider and incorporate reviewer comments in the final version.